# Improvement in Sizing Constrained Analog IC via Ts-CPD Algorithm

Pedro Lagos-Eulogio [1,†], Pedro Miranda-Romagnoli [1,*,†], Juan Carlos Seck-Tuoh-Mora [2,*,†] and Norberto Hernández-Romero [2,†]

1 Área Académica de Matemáticas y Física, Instituto de Ciencias Básicas e Ingeniería, Universidad Autónoma del Estado de Hidalgo, Carr. Pachuca-Tulancingo km. 4.5, Pachuca 42184, Hidalgo, Mexico; plagos@uaeh.edu.mx
2 Área Académica de Ingeniería, Instituto de Ciencias Básicas e Ingeniería, Universidad Autónoma del Estado de Hidalgo, Carr. Pachuca-Tulancingo km. 4.5, Pachuca 42184, Hidalgo, Mexico; nhromero@uaeh.edu.mx
* Correspondence: pmiranda@uaeh.edu.mx (P.M.-R.); jseck@uaeh.edu.mx (J.C.S.-T.-M.)
† These authors contributed equally to this work.

**Abstract:** In this work, we propose a variation of the cellular particle swarm optimization algorithm with differential evolution hybridization (CPSO-DE) to include constrained optimization, named Ts-CPD. It is implemented as a kernel of electronic design automation (EDA) tool capable of sizing circuit components considering a single-objective design with restrictions and constraints. The aim is to improve the optimization solutions in the sizing of analog circuits. To evaluate our proposal's performance, we present the design of three analog circuits: a differential amplifier, a two-stage operational amplifier (op-amp), and a folded cascode operational transconductance amplifier. Numerical simulation results indicate that Ts-CPD can find better solutions, in terms of the design objective and the accomplishment of constraints, than those reported in previous works. The Ts-CPD implementation was performed in Matlab using Ngspice and can be found on GitHub (see Data Availability Statement).

**Keywords:** cellular particle swarm optimization (CPSO); constrained optimization; circuit sizing tool; particle swarm optimization

## 1. Introduction

In recent years, analog circuit design has received much attention, particularly those with Very Large Scale of Integration (VLSI), because optimization is a process that involves many conflicting constraints and a wide range of parameters [1]. For small circuits, the equations can be stated by hand, as in the design of passive filters [2]. However, developing more robust Computer-Aided Design (CAD) and Electronic Design Automation (EDA) tools is necessary to increase productivity and quality and minimize design costs [3].

The design of analog circuits comprises three major stages: selecting a topology, sizing components, and layout extraction [4]. In the case of sizing, it is possible to use the experience when the circuits are small, but manual circuit-sizing in analog design is a time-consuming process [5]. When the circuit grows, it is impossible to size the components solely by experience; thus, mathematical tools are necessary to optimize the circuits [6].

The complexity when manually implementing an analog project is usually weeks or months. CAD and EDA tools are used to improve the design process; today's analog design environment is made of CAD tools for editing, evaluation, and design verification of analog integrated circuits, for example, HSPICE, SMASH, and CADENCE. Circuit simulators do not allow the use of methods such as quadratic or geometric programming, which exploit particular characteristics of the models. As a result, stochastic heuristic optimization techniques are used instead [7], such as in [8].

Many optimization techniques and tools for automation design have been developed over time [9,10]. Also, the fuzzy logic has been used for the circuit design as in [11,12], or in [13], where a multi-objective design is presented, while in [14], a tool for analog synthesis is introduced. In [15], a Neuro-Fuzzy method for analog circuit design is presented; it is of easy implementation, natural understanding, and better performance than static methods of fuzzy optimization; however, it still needs the human experience in the particular circuit to be designed. In [16], the application of an innovative algorithm of the type Customized Genetic Algorithm (CAG) is reported. Its purpose is to improve the optimization process of analog Complementary Metal-Oxide-Semiconductor (CMOS) ICs. A framework for facilitating the design of analog amplifiers is presented in [17].

More recently, evolutionary algorithms have been successfully applied to component value selection for analog active filters [18,19], facility location problem [20], truss structures [21], and to the analog integrated circuits design as in [22], where the sizing is achieved using a Particle Swarm Optimization (PSO) algorithm implemented in MATLAB R2008a and the results verified at the end with SPICE. In [23], a CMOS differential amplifier and a two stages CMOS op-amp are optimized to occupy the minimal possible area by the circuits and to improve their performances using the gravitational search algorithm in combination with the particle swarm optimization (GSA-PSO). The design is formulated as an optimization problem with a single objective function, although certain manual tuning is necessary to resolve conflicts with either design or performance parameters when using this method. In the work [24], a crazy PSO (CRPSO) is applied to improve the premature convergence to a local minimum of the PSO; the application optimizes the minimization of the total Metal-Oxide-Semiconductor (MOS) area of two amplifier configurations, a two-stage P-Channel MOS (PMOS) type operational amplifier, and an N-Channel MOS (NMOS) cascade code amplifier.

Heuristic techniques are necessary to solve problems with many design constraints [25]. Although they do not guarantee finding the optimal solution exactly, they provide an acceptable approximation to it in an acceptable computation time [26]. Therefore, another challenge for sizing high-performance analog circuits with tight specifications is the need for a powerful enough optimization kernel for EDA tools to handle tighter specifications and improve optimization capability [27]. Different optimization kernels are currently used for EDA tools; among them, we can mention the kernels based on GA [28], PSO [29], Ant Colony Optimization (ACO) in [30], Simulated Annealing (SA) in [31], GSA in [23], Non-dominated Sorting Genetic Algorithm-II (NSGA-II) in [32] and NSGA-II, Multi-Objective Particle Swarm Optimization (MOPSO), and Multi-Objective Simulated Annealing (MOSA) in [33].

Most heuristic methods used in the optimization kernels of the EDA tools are based on multi-objective optimization techniques [7,32] or use a restriction approach with a single objective and static penalty functions [34]. Penalty functions penalize non-feasible solutions by adding a specific value to the objective function as an amount proportional to the violation of the restriction. Thus, the optimization problem is transformed into a restrictionless optimization problem. The main problem with this methodology is choosing the appropriate penalty factor for a particular problem; it is often a complicated task, but if an adequate factor is selected, a premature convergence can occur or solutions outside the feasible region can be obtained [35]. Another approach currently used in problems with restrictions is self-adaptive penalty functions, which significantly improve the results [36]. Unfortunately, many last-generation restricted optimization methods have yet to be introduced into EDA tools. Therefore, advanced restricted optimization methods should be applied to circuit dimensioning tools to address this challenge.

In recent years, algorithms inspired by cellular automata neighborhoods to perform a local search, such as Cellular-PSO (CPSO) [37], CPSO-DE [38], Continuous-state Cellular Automata Algorithm (CCAA) [39], and Majority-minority Cellular Automata Algorithm (MmCAA) [40], have shown excellent performance in solving global optimization problems, demonstrating a good balance between exploration and exploitation, as well as a good

speed of convergence. Among them, the CPSO-DE has proven to be an excellent design method for identifying adaptive IIR systems due to the use of a differential evolution rule for the neighborhoods of cellular automata of the PSO that improves the balance between exploration and exploitation than the original version of the CPSO.

According to the previous observations, this document introduces the hybrid continuous optimization algorithm called CPSO-DE that incorporates local-search neighborhoods to improve PSO exploitation capabilities with DE exploitability. The algorithm was tested on established benchmark functions of Congress on Evolutionary Computation (CEC 2005) [41] against 7 recently published algorithms for global optimization, yielded satisfactory results.

Additionally, Deb's rules were incorporated into the algorithm to address constrained optimization [42,43]; this algorithm is called Ts-CPD applied in a single design objective problem, for the sizing of analog circuits to improve their performance. The approach is used as the optimization core of an EDA tool to size CMOS analog circuits efficiently. In particular, we focus on diminishing the total component area as the objective. At the same time, other specifications, such as dc gain, bandwidth and power dissipation, are treated as constraints that guarantee good overall performance. The circuits chosen for testing our method are well known, which allows a comparison of results with other proposals. We implemented the optimization in Matlab while the circuit simulation was done in Ngspice. Both optimization and simulation parts are linked.

We compare our proposal with previously published works, including PSO variants such as Particle Swarm Optimization (PSO) [22], Genetic Algorithm (GA) [44], Harmony Search (HS) [45], Differential Evolution (DE) [45], Artificial Bee Colony (ABC) [45], Gravitational Search Algorithm PSO (GSA-PSO) [23], Geometric Programming (GP) [46] and Aging Leader and Challenger PSO (ALC-PSO) [1]. The results show that Ts-CPSO can find a better circuit design solution than the above-listed approaches. In addition, it shows a rapid convergence in all the studied cases.

Overall, the proposed CPSO-DE algorithm is easy to understand, performs exceptionally well for continuous optimization, and is modified with Deb's rules to define the Ts-CPD algorithm in order to tackle problems with multiple constraints, as demonstrated in the area optimization of CMOS analog circuits.

The rest of the paper is organized as follows: Section 2 gives a review of CPSO-DE, while the hybridization of CPSO-DE with constrained optimization is explained in Section 3. Section 4 describes three circuits in terms of their design variables and constraints. Section 5 validates the proposed Ts-CPD through three cases of study, contrasting the findings against results from previous works. Finally, this article is concluded in Section 6.

## 2. Review of CPSO and CPSO-DE

### 2.1. Cellular Particle Swarm Optimization

PSO is one of the most frequently applied swarm intelligence-based algorithms for optimization tasks. PSO simulates the behavior of a bird flock, looking for an equilibrium between exploration and exploitation of the current solutions. Particles in a $d$-dimensional search space are regarded as candidate solutions. We denote the $i$-th particle as,

$$X_i = (x_{i,1}, x_{i,2}, \ldots, x_{i,d}), \tag{1}$$

and its velocity as,

$$V_i = (v_{i,1}, v_{i,2}, \ldots, v_{i,d}). \tag{2}$$

Each particle evolves in the search space, where $P_i = (p_{i,1}, p_{i,2}, \ldots, p_{i,d})$ is the personal best position of the $i$-th particle so far and $G = (g_1, g_2, \ldots, g_d)$ is the global best position

discovered by the swarm. At each time step $t$, both the velocity and position of each particle are updated to move it into a new position. Velocity and position are updated as follows:

$$V_i^{t+1} = V_i^t + c_1 r_1 (P_i^t - X_i^t) + c_2 r_2 (G^t - X_i^t)$$
$$X_i^{t+1} = X_i^t + V_i^t \tag{3}$$

where $c_1$ and $c_2$ are two positive constants (cognitive and social factors), $r_1$ and $r_2$ are two uniform random numbers in $[0, 1]$. The fitness $h(X_i)$ of a particle gives its quality, that is, a better fitness value means a better particle.

Several papers have presented the adaptation, modification, and hybridization of PSO with other techniques to solve a huge variety of problems. Relevant surveys can be consulted in [47–49].

CPSO is a recent variant of PSO that enhances its performance by applying a local search based on cellular automata neighborhoods [37]. In this reference, it is explained that there are two crucial factors in population-based optimization algorithms: communication mechanisms for the cooperation of the population and information inheriting for the self-adaption of each individual.

The concept of cellular automata (CAs) was first proposed by Von Neumann and Ulam, and there are an increasing number of researchers using CAs in physics, biology, social science, computer science, and so on [50–52].

CAs are discrete dynamical systems that operate on a grid of cells. Each cell initially takes a value from a finite set of states. The simplest CAs are one-dimensional, such as elementary CA (ECAs), where each cell can be in one of two states, like 0 or 1. To update the state of a cell in an ECA, the current state of the cell and its neighbors on either side are taken into account. This creates neighborhoods of three cells, and a mapping specifies how the central cell of each neighborhood evolves. This mapping, known as the ECA evolution rule, determines how each cell's state changes over time. To ensure that all cells have complete neighborhoods, periodic boundary conditions are typically applied, meaning that cells at the ends of the one-dimensional array are joined together. All cells update their state simultaneously, generating a new array of states. In the case of ECAs, each neighborhood consists of three cells, and each cell can take one of two possible values: 0 or 1. This results in a total of 8 possible neighborhoods. The central cell of each neighborhood can evolve in one of two ways, resulting in a total of 256 different possible evolution rules.

ECAs have been extensively studied due to their ability to generate a wide range of global behaviors, from fixed to complex behaviors. Figure 1 shows examples of different ECAs taking 200 cells and 200 evolutions, where the evolution rule is identified by the binary value that specifies the evolution rule, taking the mapping of the neighborhood 000 as the least significant bit. The state 0 is represented with green color and the state 1 with yellow color. These examples demonstrate the evolution towards fixed point (A), periodic (B), chaotic (C), and complex (D) behaviors.

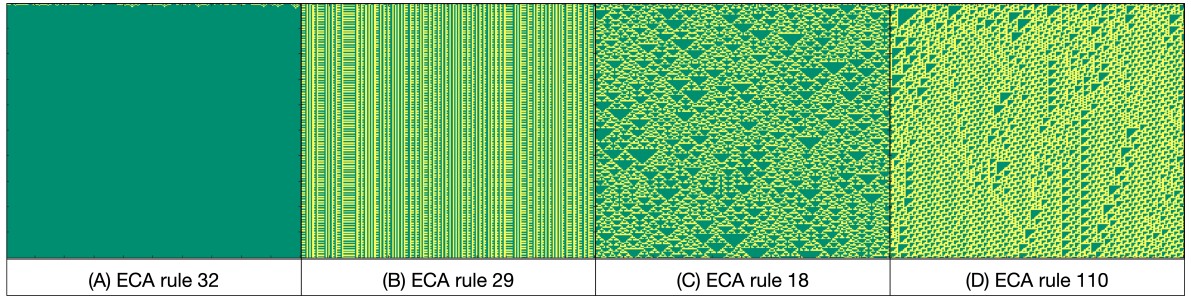

**Figure 1.** Different ECA evolution rules and the dynamic behavior observed in each of them.

CAs have the ability to generate interesting global behaviors by locally mapping blocks to individual states. In this study, we hypothesize that this CA property can be integrated as an instrument to improve the effectiveness of the PSO. Specifically, we introduce a local

search inspired by the neighborhood of a CA into the PSO operation. This mechanism enables a solution to update its position by taking information from neighboring solutions.

In this paper, we use the Cellular PSO Outer version (CPSO-outer). In this case, every particle improves its searching capability, generating new solutions not belonging to the swarm. The whole search space is considered the cell space, so every potential candidate solution in the search space can be a cell. Every particle in the swarm is a "smart-cell", defined by (1), able to construct its neighborhood by a local function, enhancing its searching capability.

The neighborhood function makes CPSO-outer differ from common PSO adopting static neighbors. Every particle $X_i$ (or "smart-cell") in CPSO-outer generates a set of $l$ neighbors $N_{i+1} \ldots N_{i+l}$ taking its current position and the global best position in order to realize a local search, following the next equation [37]:

$$
N_{i+j} = \begin{cases}
X_i^t + \frac{h(G)}{h(X_i^t)} R \circ V_i^t & h(X_i^t) \neq 0, \quad h(G) \geq 0 \\
X_i^t + \left| \frac{h(X_i^t)}{h(G)} \right| R \circ V_i^t & h(X_i^t) \neq 0, \quad h(G) < 0 \\
X_i^t + \left( \frac{e^{h(G)}}{e^{h(X_i^t)}} \right)^2 R \circ V_i^t & h(X_i^t) = 0, \quad h(G) \geq 0 \\
X_i^t + \left( \frac{e^{h(G)}}{e^{h(X_i^t)}} \right)^2 R \circ V_i^t & h(X_i^t) = 0, \quad h(G) < 0
\end{cases}
\tag{4}
$$

for $1 \leq j \leq l$. $R$ is a vector composed of $d$ uniform random numbers in $[-1, 1]$ to obtain random changes in the direction and distance of every new neighbor, and $\circ$ is the Hadamard product, $h(G)$ is the fitness of the global best position, $h(X_i^t)$ is the fitness of ith particle. The idea is that the search range of every particle would be negligible at early iterations when the difference of its fitness value with that of $h(G)$ is relatively significant. Then, when particles converge gradually to $h(G)$, a more extensive search range is used.

The neighbors generated by each particle are evaluated, and the neighbor with the best fitness value replaces the particle:

$$
f(\phi) = min(h(X_i), h(N_{i+1}), \ldots, h(N_{i+l}))
$$

$$
X_\phi = \begin{cases}
X_i & \text{if} \quad f(\phi) = h(X_i) \\
N_{i+j} & \text{if} \quad f(\phi) = h(N_{i+j})
\end{cases}
\tag{5}
$$

$$
X_i^{t+1} = X_\phi^t.
$$

This transition rule gives particles new information to explore the search space from an optimal local area to another optimal local area with better fitness value and enhance the diversity of the swarm. So CPSO-outer has more significant potential to search for the global optimum.

The CPSO has been applied and modified to solve a variety of theoretical and practical problems. For instance, in [53], CPSO is used to optimize a milling system. In [54], truss structures are optimized using variants of CPSO, and parameters controlling process planning are tuned by the application of CPSO [55]. Nevertheless, CPSO has not been implemented for sizing analog circuit components.

### 2.2. Hybrid Cellular Particle Swarm Optimization and Differential Evolution

Hybrid cellular particle swarm optimization and differential evolution (CPSO-DE) is a recent hybrid method that combines the features of PSO, CA, and DE [38], which is an incorporation of local differential search to the CPSO-outer algorithm.

The CPSO-DE algorithm utilizes local differential-search elements, which can be defined as follows.

(a)  configuration: ($Q$ particles or smart-cells);

(b)  cell space: the set of all cells;

(c)  cell state: the particle's information at time $t$, $S_i^t = [X_i^t]$;

(d)  neighborhood: $\Phi(i) = \{i + \delta_j\}, 1 \leq j \leq l$ ($l$ is the neighborhood size). See Figure 2,

(e)  transition rule: $S_i^{t+1} = \varphi(S_i^t \cup S_{\Phi(i)}^t)$.

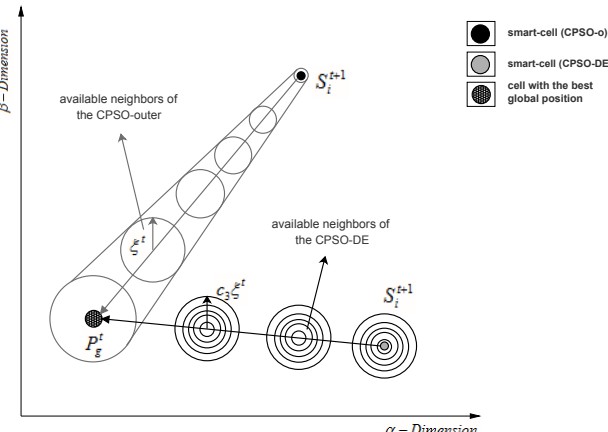

**Figure 2.** Neighborhood for CPSO-outer and CPSO-DE.

In CPSO-DE, the $i$-th cell state $S_i^t$ in the iteration $t$ is updated using the PSO algorithm as follows:

$$V_i^{t+1} = w^t V_i^t + c_1 r_1 (P_i^t - S_i^t) + c_2 r_2 (P_g^t - S_i^t) \tag{6}$$

$$S_i^{t+1} = S_i^t + V_i^{t+1} \tag{7}$$

where $i = 1, 2, \ldots, Q$ is the cell index and $Q$ is the number of smart cells, $c_1$ and $c_2$ are the cognitive and social acceleration parameters respectively, $r_1$ and $r_2$ are two uniform distributed random numbers within $[0, 1]$, $w$ is the inertial weight and decreases linearly. $P_i$ is the previous personal best position, $P_g$ is the global best position, and $X_i$ and $V_i$ are the current positions and velocity.

The operators used to determine each smart cell's neighborhoods are mutation and crossover. The mutation scheme "DE/rand/1" creates a new solution as follows:

$$O_{i,k}^t = S_{r_1}^t + c_3 (S_{r_2}^t - S_{r_3}^t) \tag{8}$$

where $k = 1, 2, \ldots, l$ enumerates every neighbor, and $l$ is the neighborhood size. The $r_1, r_2, r_3 \in \{1, 2, \ldots, Q\}$ are randomly chosen integers, distinct from each other and different from $i$. Factor $c_3$ is a real value between $[0, 2]$ for scaling the difference vector.

The crossover is an introduction to creating $l$ trial vector $H_{i,k}$, combining the information of the current smart cell with each one of the $l$ mutated vectors, as follows:

$$H_{i,j,k} = \begin{cases} O_{i,j,k'}^t & \text{if } r_{i,j} \leq C_r \text{ or } j = j_{rand}, \\ \\ S_{i,j'}^t & \text{otherwise,} \end{cases} \tag{9}$$

where $r_{ij}$ is a uniformly distributed random number within $[0, 1]$, $C_r \in [0, 1]$ is the crossover probability factor, and $j_{rand} \in \{1, 2, \ldots, D\}$ is a randomly chosen index, which ensures that $H_{i,k}$ copies at least one component from $O_{i,k}$. Finally, the transition rule is applied over the trial vectors to update the state of the current smart-cell:

$$S_i^{t+1}(P_\Phi) = \varphi(f(S_i^{t+1}), f(H_{i,1}), f(H_{i,2}), \ldots, f(H_{i,l})) \tag{10}$$

where the $f(.)$ are the fitness functions. In CPSO-outer, the neighborhood function $\Phi(i)$ generates random neighbors within radius $\xi^t$ away from $S_i^t$ according to its fitness value

and the fitness of the best particle. Radius $\xi^t$ is small when the smart-cell $S_i^t$ is far from $P_g^t$, so the potential neighbors are close to $S_i^t$, and only when $S_i^t$ converges to an equilibrium point, $\xi^t$ would be a uniform random number in $[-1, 1]$. Therefore, the radius of neighborhoods in CPSO-outer increases when the particles stabilize. Therefore, the best results are obtained up to the last iterations.

On the other hand, CPSO-DE generates a random neighbor within radius $\xi^t = c_3(S_{r2}^t - S_{r3}^t)$. Thus, the radius of neighborhoods depends on the distribution and the improved information of the swarm as iteration passes, not just from the difference with the best global position. Thus, $S_i^t$ is more likely to obtain better neighbors in any iteration time.

## 3. Tournament-Selection CPD

The use of local search strategies inspired by cellular automata neighborhoods in heuristic algorithms has been shown to be effective, more specifically, in the use of adaptive IIR filters through the hybridization of the CPSO and DE algorithms that use a rule based on the use of neighborhoods. However, the CPSO-DE algorithm for the problem with restrictions on the sizing of CMOS circuits has not yet been reported in the literature, hence the motivation for this work.

In this section, we explain the parts that comprise the proposed Ts-CPD algorithm. First, we describe the optimization problem to be solved, which contemplates restrictions. Next, we explain how the initial values are selected, for our algorithm, using tournament selection (Ts), which is a variant of what Deb proposed [42]. We conclude this section by explaining the implementation of the Deb rule in the CPSO-DE, to build the new Ts-CPD algorithm.

### 3.1. The Circuit Design Problem

Many optimization problems in science and engineering implicate some constraints that the optimal solution must satisfy. For example, in a generic circuit, the optimization problem consists of finding optimal values of the design parameters. Then, a circuit design problem is usually written as a nonlinear programming (NLP) problem of the following type:

$$
\begin{aligned}
&\text{minimize } f(X) \\
&\quad X \in \mathbb{R}^n
\end{aligned}
$$

Subject to:

$$
\begin{array}{ll}
|g_p| \geq spec_{g_p} & p = 1 \cdots r \\
h_q = spec_{h_q} & q = 1 \cdots s \\
x_{i,min} \leq x_i \leq x_{i,max} & i = 1 \cdots n
\end{array}
\tag{11}
$$

In the above NLP problem, $f$ is the cost function that maps the input space into the output one, $f : \mathbb{R}^n \to \mathbb{R}$, with $n = k + m$. There are two types of constraints, inequality constraints $g_p$ that have to be major or minor than certain $spec_{g_p}$, and the equality constraints $h_q$, that has to be equal to the restriction $spec_{h_q}$. The $i$th variable varies in the range $[x_{i,min}, x_{i,max}]$.

The $k$ independent variables and $m$ dependent ones determine the circuit design represented in a single vector as,

$$
X = (x_1, \ldots, x_k, x_{k+1}, \ldots, x_{k+m}).
\tag{12}
$$

The design variables and constraints for specific circuits studied in this paper are given in the Section 4.

### 3.2. Tournament-Selection

As in the cost function f(X) of the optimization problem expressed in (11), the restrictions are not considered; we need a method that allows us to assess their contribution. In [42], Deb proposes a constraint handling method so that while the cost function is mini-

mized, the constraints in the search for the minimum are considered. We will use Deb's method in this work, as explained below.

Let's say that the CPSO-DE algorithm has encountered two solutions for the problem (11), $X_1$ and $X_2$, according to the constrained optimization, solution $X_1$ is considered better if [43]:

1. both solutions are feasible, but $X_1$ cost $\leq X_2$ cost; or,
2. $X_1$ is feasible but $X_2$ is not; or,
3. both solutions are unfeasible, but $X_1$ has less overall constraint violations than $X_2$.

These rules, implemented as Algorithm 1, are advantageous in finding a better solution for the circuit design, as will be shown in Section 5.

---

**Algorithm 1** Tournament-Selection $(X_1, X_2)$

---

1: **if** $X_1$ is feasible **and** $X_2$ is feasible **then**
2: 　　**if** $f(X_1) < f(X_2)$ **then**
3: 　　　　**return** $(X_1)$
4: 　　**else**
5: 　　　　**return** $(X_2)$
6: 　　**end if**
7: **else if** constraints violation $(X_1) <$ constraints violation $(X_2)$ **then**
8: 　　**return** $(X_1)$
9: **else**
10: 　　**return** $(X_2)$
11: **end if**

---

### 3.3. Ts-CPD Algorithm

This work proposes a new methodology that combines the CPSO-DE algorithm and Deb's rules for the problem of sizing CMOS analog circuits with constraints. The proposed algorithm, Ts-CPD, incorporates the tournament selection (see Algorithm 1) in the $\Psi()$ function. In this method, a new transition rule is proposed for Ts-CPD, which is applied to the trial vectors to update the state of the current smart-cell:

$$S_i^{t+1}(P_\Phi) = \Psi(\ldots\Psi(\Psi(S_i^{t+1}, H_{i,1}), H_{i,2}), \ldots, H_{i,l}) \tag{13}$$

The transition rule in (13) means that each cell in the neighborhood (including the same smart-cell) competes in a paired tournament (according to Deb criteria), and the winner is chosen to update the state of the smart-cell.

The proposed Ts-CPD method is described in Algorithm 2. First, the algorithm sets the control parameters $Q$, $l$, $T$, $x_{min}$, $x_{max}$ y $v_{max}$. Next, the state $(S)$ and velocity $(V)$ are randomly initialized for each smart-cell. Then, each cell is evaluated, and its number of violated constraints is quantified. In line 9, Algorithm 1 is used to identify the best global position. The process halts according to the stopping criteria of iteration and convergence, according to line 10. Then, the cell state is updated using (6) and (7) in line 12. Later, the neighborhood of size $l$ is generated for each smart-cell, using the DE method. Each neighbor is defined by the mutation and crossover rules in lines 14 and 15 using (8) and (9), respectively. The new transition rule inspired by Deb's rules and CA behavior, defined in (13), is applied in line 16 to determine the new cell state. Finally, the best local and global positions are updated in lines 18 and 19, respectively, using Algorithm 1. The process is repeated by each smart-cell and neighbor.

---

**Algorithm 2** Ts-CPD

---

1: **//\*\* Initialization**
2: Set the control parameters: $Q$, $l$, $T$, $x_{min}$, $x_{max}$, $v_{max}$;
3: **for** $i = 1$ to $Q$ **do**
4:　　Initialize $S_i \in (x_{min}, x_{max})$ randomly;
5:　　Initialize $V_i \in (-v_{max}, v_{max})$ randomly;
6:　　$P_i = S_i$;
7: **end for**
8: Evaluate each cell $f(S_i)$;
9: Identify the best global position ($P_g$): using Algorithm 1;
　　**//\*\*\* Loop**
10: **while** stopping criterion is not satisfied **do**
11:　　**for** $i = 1$ to $Q$ **do**
12:　　　　**Update cell state:** using Equations (6) and (7);
　　　　　**//\*\*\*Generate l neighbors using DE method**
13:　　　　**for** $k = 1$ to $l$ **do**
14:　　　　　　**Mutation rule:** using Equation (8);
15:　　　　　　**Crossover rule:** using Equation (9);
16:　　　　　　**New transition rule:** using Equation (13);
17:　　　　**end for**
18:　　　　Identify the best local position ($P_i$): using Algorithm 1;
19:　　**end for**
20:　　Identify the best global position ($P_g$): using Algorithm 1;
21: **end while**

---

### 3.4. Performance of the Ts-CPD Algorithm

To test the effectiveness of the Ts-CPD algorithm (without Deb's rules), we compared it to seven recently published algorithms, namely Archimedes Optimization Algorithm (AOA) [56], Harris Hawks Optimization (HHO) [57], Weighted Superposition Attraction (WSA) [58], CCAA [39], MmCAA [40], Reversible Elementary Cellular Automata (RECAA) [59], and Political Optimizer (PO) [60]. 25 benchmark functions were used from CEC 2005 benchmark functions [41], which included five unimodal functions ($f_1, \ldots, f_5$), seven multimodal functions ($f_6, \ldots, f_{12}$), two expanded multimodal functions ($f_{13}, f_{14}$), and 11 hybrid composition multimodal functions ($f_{15}, \ldots, f_{25}$). We obtained the codes and parameters for these algorithms from the references cited in this study. This ensured that we used the same implementations as the original authors, making the comparison more objective. All parameter settings are given in Table 1.

**Table 1.** Parameter settings of algorithms employed for comparison with Ts-CPD.

| Algorithm | Parameters |
|---|---|
| Ts-CPD | $Q = 12$, $l = 6$, $c_1 = 2$, $c_2 = 2$, $c_3 = 0.5$, $C_r = 0.9$, $w_{max} = 0.7$, $w_{min} = 0.15$ |
| AOA | $N = 60$, $C_1 = 2$, $C_2 = 6$, $C_3 = 1$, $C_4 = 0.2$, $u = 0.9$, $l = 0.1$ |
| HHO | $N = 60$, $E_0 = 2 \cdot rand - 1$, $J = 2 \cdot (1 - rand)$ |
| WSA | $N = 60$, $\tau = 0.8$, $sl_o = 0.035$, $\lambda = 0.75$, $\varphi = 0.001$ |
| CCAA | $Q = 12$, $l = 6$, $p_h = 2$, $p_l = 1$, $d_M = 1$, $d_m = 3$, $r_{max} = 4$, $r_{min} = 1$ |
| MmCAAA | $Q = 12$, $l = 6$, $d_M = 1.7$, $r_{max} = 6$, $r_{min} = 2$, $e = 2$ |
| RECAA | $Q = 12$, $l = 6$, $p = 1.5$, $r_{max} = 6$, $r_{min} = 1$ |
| PO | $n = 8$ ($N = 64$), $\lambda = 1$, $areas = 7$, $parties = 7$ |

Tables 2 and 3 present the average values and standard deviations of the objective function values obtained by each algorithm. We ran each algorithm independently 30 times. In unimodal problems, the Ts-CPD algorithm showed excellent performance, ranking

first among the eight algorithms in terms of average value. Moreover, it surpassed other algorithms in three cases based on standard deviation, highlighting its proficiency in information exploitation.

**Table 2.** Performance of metaheuristic algorithms compared with Ts-CPD on 30-dimensional unimodal problems. The best values are in bold.

| Benchmark | | Ts-CPD | AOA | HHO | WSA | CCAA | MmCAA | RECAA | PO |
|---|---|---|---|---|---|---|---|---|---|
| $f_1$ | Avg | **1.08** | $3.54 \times 10^4$ | $2.51 \times 10^3$ | $7.42 \times 10^4$ | $1.99 \times 10^4$ | $2.23 \times 10^3$ | $8.51 \times 10^2$ | $1.88 \times 10^4$ |
| | Std | **2.93** | $5.35 \times 10^3$ | $1.22 \times 10^3$ | $5.97 \times 10^3$ | $7.33 \times 10^3$ | $6.25 \times 10^2$ | $4.39 \times 10^2$ | $7.28 \times 10^3$ |
| $f_2$ | Avg | **$1.60 \times 10^2$** | $4.11 \times 10^4$ | $2.40 \times 10^4$ | $1.11 \times 10^5$ | $2.44 \times 10^4$ | $4.23 \times 10^4$ | $1.70 \times 10^4$ | $1.92 \times 10^4$ |
| | Std | **$1.68 \times 10^2$** | $5.24 \times 10^3$ | $2.82 \times 10^3$ | $3.02 \times 10^4$ | $4.56 \times 10^3$ | $6.21 \times 10^3$ | $5.52 \times 10^3$ | $4.97 \times 10^3$ |
| $f_3$ | Avg | **$2.60 \times 10^6$** | $5.23 \times 10^8$ | $1.11 \times 10^8$ | $1.00 \times 10^8$ | $1.19 \times 10^8$ | $7.82 \times 10^7$ | $3.79 \times 10^7$ | $1.68 \times 10^8$ |
| | Std | $1.40 \times 10^6$ | $1.74 \times 10^8$ | $3.41 \times 10^7$ | **0.00** | $5.26 \times 10^7$ | $2.80 \times 10^7$ | $1.33 \times 10^7$ | $1.05 \times 10^8$ |
| $f_4$ | Avg | **$9.37 \times 10^2$** | $4.83 \times 10^4$ | $5.46 \times 10^4$ | $1.11 \times 10^5$ | $3.49 \times 10^4$ | $5.26 \times 10^4$ | $2.69 \times 10^4$ | $2.89 \times 10^4$ |
| | Std | **$4.80 \times 10^2$** | $7.16 \times 10^3$ | $8.20 \times 10^3$ | $2.13 \times 10^4$ | $6.61 \times 10^3$ | $7.99 \times 10^3$ | $5.06 \times 10^3$ | $6.62 \times 10^3$ |
| $f_5$ | Avg | **$5.78 \times 10^3$** | $2.97 \times 10^4$ | $2.49 \times 10^4$ | $4.32 \times 10^4$ | $2.17 \times 10^4$ | $1.41 \times 10^4$ | $8.76 \times 10^3$ | $2.52 \times 10^4$ |
| | Std | $1.57 \times 10^3$ | $3.48 \times 10^3$ | $3.38 \times 10^3$ | $4.19 \times 10^3$ | $3.87 \times 10^3$ | $2.32 \times 10^3$ | **$1.24 \times 10^3$** | $2.73 \times 10^3$ |

**Table 3.** Performance of metaheuristic algorithms compared with Ts-CPD on 30-dimensional multimodal problems. The best values are in bold.

| Benchmark | | Ts-CPD | AOA | HHO | WSA | CCAA | MmCAA | RECAA | PO |
|---|---|---|---|---|---|---|---|---|---|
| $f_6$ | Avg | **$2.40 \times 10^3$** | $1.10 \times 10^{10}$ | $1.32 \times 10^8$ | $1.00 \times 10^8$ | $2.56 \times 10^9$ | $4.27 \times 10^7$ | $2.00 \times 10^6$ | $2.98 \times 10^9$ |
| | Std | $4.84 \times 10^3$ | $3.16 \times 10^9$ | $1.20 \times 10^8$ | **0.00** | $1.93 \times 10^9$ | $1.67 \times 10^7$ | $1.19 \times 10^6$ | $1.98 \times 10^9$ |
| $f_7$ | Avg | **4.72** | $1.32 \times 10^3$ | $3.15 \times 10^2$ | $3.61 \times 10^3$ | $5.85 \times 10^1$ | $1.40 \times 10^2$ | $5.57 \times 10^1$ | $4.81 \times 10^1$ |
| | Std | **7.72** | $1.68 \times 10^2$ | $7.29 \times 10^1$ | $5.20 \times 10^2$ | $1.47 \times 10^1$ | $3.44 \times 10^1$ | $1.70 \times 10^1$ | $2.70 \times 10^1$ |
| $f_8$ | Avg | $2.10 \times 10^1$ | $2.11 \times 10^1$ | $2.09 \times 10^1$ | $2.11 \times 10^1$ | $2.10 \times 10^1$ | $2.10 \times 10^1$ | $2.10 \times 10^1$ | **$2.05 \times 10^1$** |
| | Std | **$5.48 \times 10^{-2}$** | $8.26 \times 10^{-2}$ | $8.58 \times 10^{-2}$ | $5.72 \times 10^{-2}$ | $8.31 \times 10^{-2}$ | $6.76 \times 10^{-2}$ | $6.42 \times 10^{-2}$ | $7.97 \times 10^{-2}$ |
| $f_9$ | Avg | **$1.02 \times 10^2$** | $3.04 \times 10^2$ | $3.14 \times 10^2$ | $4.01 \times 10^2$ | $2.24 \times 10^2$ | $2.31 \times 10^2$ | $1.63 \times 10^2$ | $2.50 \times 10^2$ |
| | Std | $2.64 \times 10^1$ | $2.18 \times 10^1$ | $2.60 \times 10^1$ | **$1.34 \times 10^1$** | $2.52 \times 10^1$ | $1.70 \times 10^1$ | $2.06 \times 10^1$ | $4.29 \times 10^1$ |
| $f_{10}$ | Avg | **$4.89 \times 10^1$** | $4.66 \times 10^2$ | $4.21 \times 10^2$ | $7.25 \times 10^2$ | $5.18 \times 10^2$ | $3.93 \times 10^2$ | $2.54 \times 10^2$ | $4.05 \times 10^2$ |
| | Std | **$1.45 \times 10^1$** | $4.68 \times 10^1$ | $7.25 \times 10^1$ | $5.27 \times 10^1$ | $6.17 \times 10^1$ | $5.68 \times 10^1$ | $2.38 \times 10^1$ | $4.22 \times 10^1$ |
| $f_{11}$ | Avg | **$2.16 \times 10^1$** | $4.17 \times 10^1$ | $4.07 \times 10^1$ | $4.37 \times 10^1$ | $2.97 \times 10^1$ | $3.55 \times 10^1$ | $3.27 \times 10^1$ | $3.04 \times 10^1$ |
| | Std | 2.57 | 2.49 | 2.56 | 2.12 | 2.07 | 2.01 | **1.41** | 8.22 |
| $f_{12}$ | Avg | $1.29 \times 10^6$ | $1.50 \times 10^6$ | $7.22 \times 10^5$ | $1.50 \times 10^6$ | $7.89 \times 10^5$ | **$6.94 \times 10^5$** | $7.14 \times 10^5$ | $9.86 \times 10^5$ |
| | Std | $1.44 \times 10^5$ | $2.46 \times 10^5$ | $2.00 \times 10^5$ | $1.66 \times 10^5$ | $1.39 \times 10^5$ | $1.67 \times 10^5$ | $1.25 \times 10^5$ | **$7.79 \times 10^3$** |
| $f_{13}$ | Avg | **6.45** | $2.82 \times 10^1$ | $3.31 \times 10^1$ | $1.27 \times 10^2$ | $1.46 \times 10^1$ | $2.20 \times 10^1$ | $1.76 \times 10^1$ | $1.15 \times 10^1$ |
| | Std | 4.29 | 5.82 | 5.18 | $3.60 \times 10^1$ | 2.80 | **1.40** | 1.72 | 3.09 |
| $f_{14}$ | Avg | **$1.30 \times 10^1$** | $1.35 \times 10^1$ | $1.37 \times 10^1$ | $1.39 \times 10^1$ | $1.34 \times 10^1$ | $1.36 \times 10^1$ | $1.34 \times 10^1$ | $1.41 \times 10^1$ |
| | Std | $3.26 \times 10^{-1}$ | $3.42 \times 10^{-1}$ | $2.04 \times 10^{-1}$ | $2.21 \times 10^{-1}$ | $2.90 \times 10^{-1}$ | **$1.35 \times 10^{-1}$** | $2.26 \times 10^{-1}$ | $1.43 \times 10^{-1}$ |
| $f_{15}$ | Avg | $5.69 \times 10^2$ | $9.80 \times 10^2$ | $7.41 \times 10^2$ | $1.24 \times 10^3$ | $6.92 \times 10^2$ | $5.69 \times 10^2$ | **$5.13 \times 10^2$** | $1.03 \times 10^3$ |
| | Std | $1.31 \times 10^2$ | $7.26 \times 10^1$ | $1.26 \times 10^2$ | $8.68 \times 10^1$ | $1.08 \times 10^2$ | **$4.77 \times 10^1$** | $5.78 \times 10^1$ | $1.16 \times 10^2$ |
| $f_{16}$ | Avg | **$2.91 \times 10^2$** | $8.22 \times 10^2$ | $4.92 \times 10^2$ | $1.17 \times 10^3$ | $5.57 \times 10^2$ | $4.04 \times 10^2$ | $3.07 \times 10^2$ | $6.98 \times 10^2$ |
| | Std | $1.79 \times 10^2$ | $9.05 \times 10^1$ | $8.22 \times 10^1$ | $1.41 \times 10^2$ | $1.05 \times 10^2$ | $5.04 \times 10^1$ | **$4.74 \times 10^1$** | $1.03 \times 10^2$ |
| $f_{17}$ | Avg | **$2.90 \times 10^2$** | $8.79 \times 10^2$ | $5.91 \times 10^2$ | $1.18 \times 10^3$ | $6.23 \times 10^2$ | $4.83 \times 10^2$ | $3.52 \times 10^2$ | $7.86 \times 10^2$ |
| | Std | $1.92 \times 10^2$ | $1.40 \times 10^2$ | $7.63 \times 10^1$ | $1.61 \times 10^2$ | $1.21 \times 10^2$ | $6.79 \times 10^1$ | **$5.10 \times 10^1$** | $9.67 \times 10^1$ |
| $f_{18}$ | Avg | $9.77 \times 10^2$ | $9.77 \times 10^2$ | **$9.00 \times 10^2$** | $9.00 \times 10^2$ | **$9.00 \times 10^2$** | **$9.00 \times 10^2$** | **$9.00 \times 10^2$** | **$9.00 \times 10^2$** |
| | Std | $6.41 \times 10^1$ | $1.44 \times 10^2$ | **0.00** | $3.95 \times 10^{-6}$ | **0.00** | **0.00** | **0.00** | **0.00** |
| $f_{19}$ | Avg | $9.72 \times 10^2$ | $1.01 \times 10^3$ | **$9.00 \times 10^2$** | $9.00 \times 10^2$ | **$9.00 \times 10^2$** | **$9.00 \times 10^2$** | **$9.00 \times 10^2$** | **$9.00 \times 10^2$** |
| | Std | $3.42 \times 10^1$ | $1.53 \times 10^2$ | **0.00** | $4.51 \times 10^{-6}$ | **0.00** | **0.00** | **0.00** | **0.00** |
| $f_{20}$ | Avg | $9.79 \times 10^2$ | $9.73 \times 10^2$ | **$9.00 \times 10^2$** | $9.00 \times 10^2$ | **$9.00 \times 10^2$** | **$9.00 \times 10^2$** | **$9.00 \times 10^2$** | **$9.00 \times 10^2$** |
| | Std | $3.99 \times 10^1$ | $1.36 \times 10^2$ | **0.00** | $4.48 \times 10^{-6}$ | **0.00** | **0.00** | **0.00** | **0.00** |
| $f_{21}$ | Avg | $9.64 \times 10^2$ | $1.31 \times 10^3$ | $1.21 \times 10^3$ | $1.40 \times 10^3$ | $1.33 \times 10^3$ | $1.25 \times 10^3$ | **$8.36 \times 10^2$** | $1.14 \times 10^3$ |
| | Std | $3.34 \times 10^2$ | $1.57 \times 10^1$ | $1.09 \times 10^2$ | **$1.18 \times 10^1$** | $3.48 \times 10^1$ | $7.07 \times 10^1$ | $1.32 \times 10^2$ | $1.95 \times 10^1$ |
| $f_{22}$ | Avg | **$9.21 \times 10^2$** | $1.42 \times 10^3$ | $1.28 \times 10^3$ | $1.82 \times 10^3$ | $1.24 \times 10^3$ | $1.13 \times 10^3$ | $1.05 \times 10^3$ | $1.09 \times 10^3$ |
| | Std | **$1.66 \times 10^1$** | $7.23 \times 10^1$ | $1.20 \times 10^2$ | $1.05 \times 10^2$ | $9.86 \times 10^1$ | $3.48 \times 10^1$ | $4.37 \times 10^1$ | $8.81 \times 10^1$ |

**Table 3.** *Cont.*

| Benchmark | | Ts-CPD | AOA | HHO | WSA | CCAA | MmCAA | RECAA | PO |
|---|---|---|---|---|---|---|---|---|---|
| $f_{23}$ | Avg | $1.07 \times 10^3$ | $1.31 \times 10^3$ | $1.24 \times 10^3$ | $1.40 \times 10^3$ | $1.35 \times 10^3$ | $1.25 \times 10^3$ | $\mathbf{9.26 \times 10^2}$ | $1.15 \times 10^3$ |
| | Std | $2.02 \times 10^2$ | $1.69 \times 10^1$ | $8.21 \times 10^1$ | $\mathbf{1.39 \times 10^1}$ | $2.77 \times 10^1$ | $4.86 \times 10^1$ | $1.66 \times 10^2$ | $2.41 \times 10^1$ |
| $f_{24}$ | Avg | $\mathbf{2.26 \times 10^2}$ | $1.37 \times 10^3$ | $1.34 \times 10^3$ | $1.46 \times 10^3$ | $1.39 \times 10^3$ | $1.33 \times 10^3$ | $9.69 \times 10^2$ | $1.07 \times 10^3$ |
| | Std | $6.48 \times 10^1$ | $1.82 \times 10^1$ | $7.33 \times 10^1$ | $\mathbf{1.39 \times 10^1}$ | $4.43 \times 10^1$ | $4.04 \times 10^1$ | $1.90 \times 10^2$ | $1.67 \times 10^2$ |
| $f_{25}$ | Avg | $\mathbf{1.01 \times 10^3}$ | $1.38 \times 10^3$ | $1.40 \times 10^3$ | $1.47 \times 10^3$ | $1.40 \times 10^3$ | $1.39 \times 10^3$ | $1.23 \times 10^3$ | $1.28 \times 10^3$ |
| | Std | $9.32$ | $2.66 \times 10^1$ | $2.99 \times 10^1$ | $\mathbf{8.30}$ | $3.26 \times 10^1$ | $1.90 \times 10^1$ | $4.54 \times 10^1$ | $1.26 \times 10^2$ |

For the 20 multimodal and hybrid problems, Ts-CPD exhibited the highest average values in 12 instances. It also demonstrated its ability to explore and exploit simultaneously while maintaining robustness, achieving the best standard deviation values in four cases.

Table 4 presents the results of the Wilcoxon rank-sum statistical test which compares Ts-CPD with other methods for each benchmark function. The symbol + indicates a better result that is statistically significant, ≈ indicates no significant difference, and − indicates a worse statistically significant result. The Avg column presents the average rank obtained by each algorithm when optimizing the benchmark functions. The Rank column shows the order in which each algorithm is ranked based on its average. Ts-CDP obtained the best rank, followed by RECAA. In all cases, Ts-CPD obtained a more significant difference in terms of the number of functions with a better significant result in this experiment. In addition, Figure 3 shows some examples of the convergence curves for different test functions in 30 dimensions.

**Table 4.** Wilcoxon rank-sum test and ranking of the compared algorithms on 30-dimensional problems.

| Algorithm | +/ − / ≈ | Avg | Rank |
|---|---|---|---|
| Ts-CPD | −/−/− | 1.88 | 1 |
| AOA | 21/0/4 | 6.04 | 7 |
| HHO | 16/3/6 | 4.52 | 6 |
| WSA | 20/2/3 | 6.88 | 8 |
| CCAA | 18/3/4 | 4.36 | 5 |
| MmCAA | 15/3/7 | 3.64 | 3 |
| RECAA | 13/3/9 | 2.12 | 2 |
| PO | 17/3/5 | 3.92 | 4 |

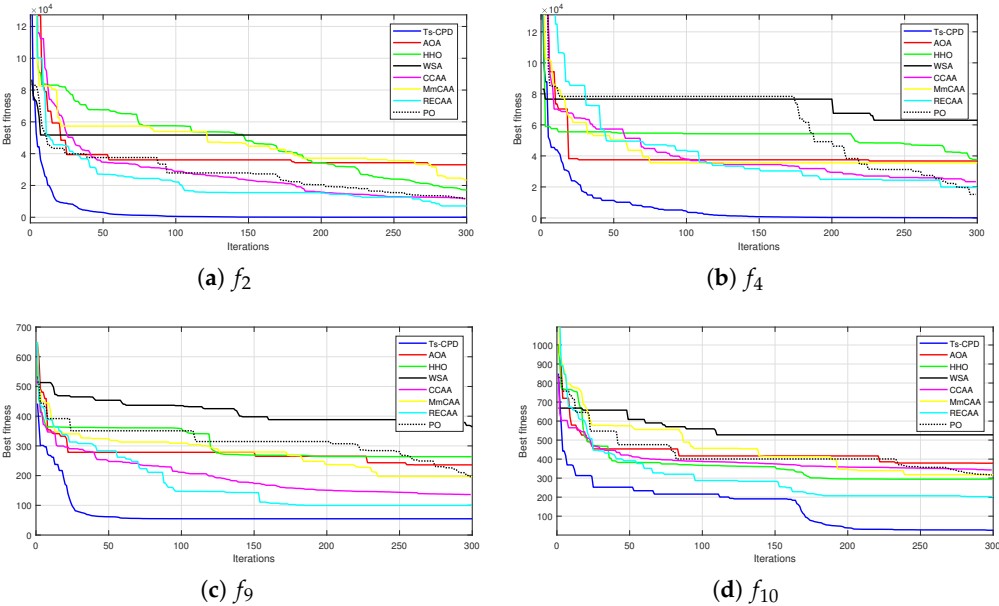

**(a)** $f_2$      **(b)** $f_4$

**(c)** $f_9$      **(d)** $f_{10}$

**Figure 3.** *Cont.*

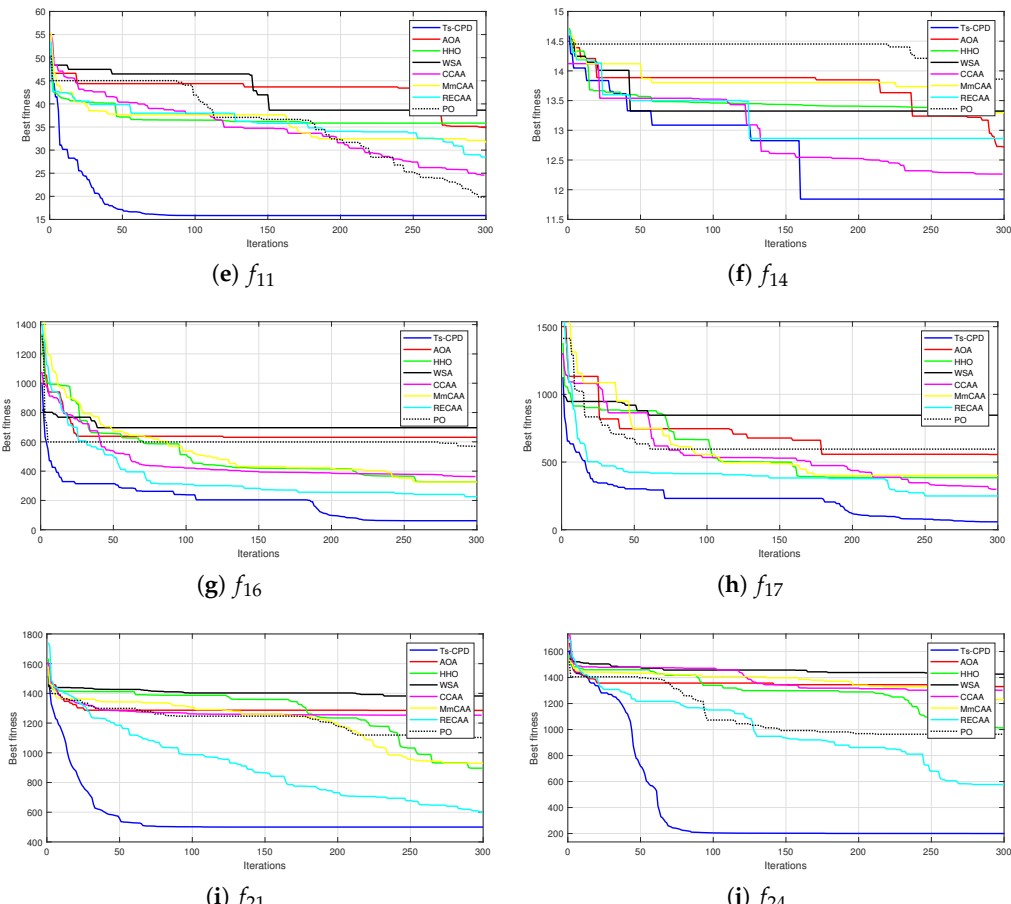

**Figure 3.** Convergence curves of the different algorithms for CEC05 functions in 30 dimensions.

### 3.5. Complexity Analysis of the Ts-CPD Algorithm

Most evolutionary algorithms imply a complexity of the following three main parts [60,61]:

1.  Initialization of population, generally bounded by $\mathbb{O}(UD)$ where $U$ is the population size and $D$ the dimensionality of the problem.
2.  Fitness evaluation is bounded in general by $\mathbb{O}(UC_{obj})$ where $C_{obj}$ is the cost of evaluating the objective function.
3.  Optimization loop, generally bounded by $\mathbb{O}(TUD + TUC_{obj})$, here $T$ is the total iteration number of the loop.

The complexity analysis of the Ts-CPD algorithm takes into account these three parts:

1.  Initialization of population is bounded by $\mathbb{O}(QD)$, similar to other algorithms (lines 3–7 in Algorithm 2).
2.  Fitness evaluation is bounded by $\mathbb{O}(QC_{obj})$ in line 8. Notice that Algorithm 1 is linear with regard to $C_{obj}$ when using Deb criteria. The best global position is calculated in $\mathbb{O}(Q)$ in line 9.
3.  For the optimization loop, smart-cells are updated with complexity $\mathbb{O}(TQD)$ (line 12); mutation and crossover have complexity $\mathbb{O}(TQlD)$ (lines 14 and 15), and the new transition rule is $\mathbb{O}(TQlC_{obj})$ in line 16. The best local position in line 18 is calculated in $\mathbb{O}(TQl)$, and the best global position is $\mathbb{O}(TQ)$ in line 20. Therefore, the complexity of the optimization loop asymptotically tends to $\mathbb{O}(TQlD + TQlC_{obj})$, which is also equivalent to the other algorithms.

The complexity analysis concludes that the Ts-CPD algorithm is asymptotically equivalent to the other state-of-the-art methods when $Ql$ is similar to $U$.

## 4. The Proposed Tool for Analog IC Sizing

The EDA tool proposed for the designer of analog circuits through the Ts-CPD algorithm allows obtaining a minimum area of the components used while complying with the design specifications. It is handy for designing the frequency response of circuits, such as bandwidth, phase margin, Common Mode Rejection Ratio (CMRR), or Power Supply Rejection Ratio (PSRR); only the slew rate can be designed in the time domain. For this purpose, before beginning the design, the designer must introduce the specifications (restrictions) of the circuit and the acceptable ranges and values for the parameters according to the technology used. The parameters to choose are the width and length of the CMOS transistors, capacitance and resistance (if any) values, bias current, and voltage sources.

The tool consists of two main modules: the optimization and synthesis processes. The optimization process contains the Ts-CPD algorithm comprising the CPSO-DE and the Deb rule, with a new transition rule given by (13); this module is implemented in Matlab. The synthesis process uses the specialized Ngspice v26 software, which allows analog circuit simulations without mathematical equations. Instead, the standard configurations necessary to evaluate the performance of circuits are implemented in a netlist format. Both modules, the optimization and synthesis processes, are linked, allowing an automatic circuit design. The flow chart for our EDA tool, using Ts-CPD, is shown in Figure 4.

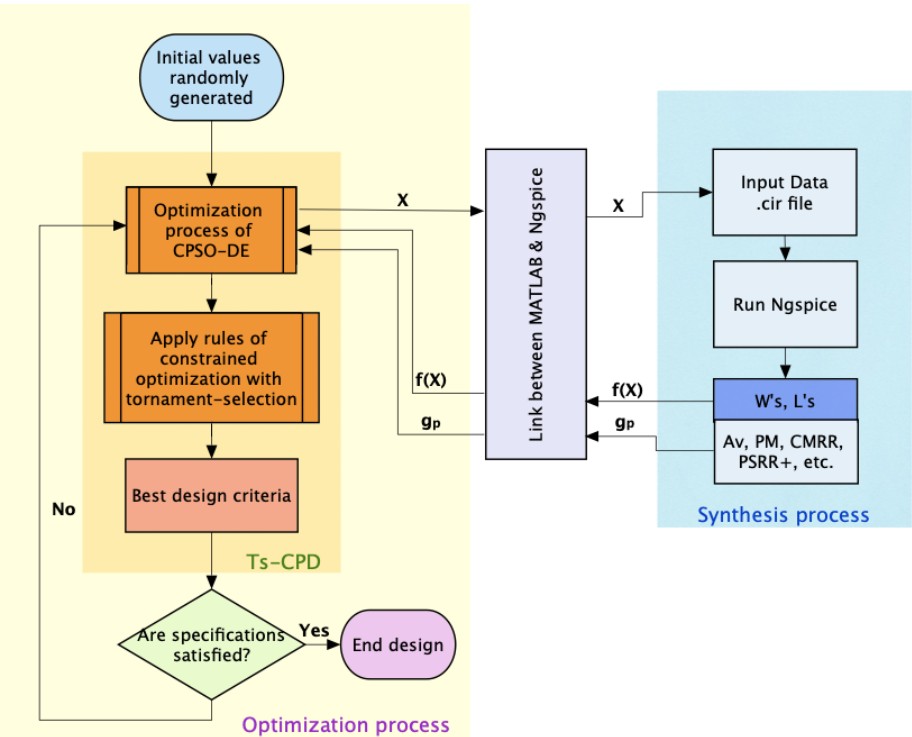

**Figure 4.** Flow chart of Ts-CPD as part of an EDA Tool.

The following subsection describes three case studies, in terms of their variables and constraints, that will be used to verify the efficiency of the EDA tool.

### 4.1. Cases of Study

To test our algorithm and tool, we chose three case studies, a "CMOS Differential Amplifier", a "CMOS two-stage operational amplifier", and a "CMOS folded cascode operational transconductance amplifier". These cases were chosen because they have already been studied previously, and therefore, it is possible to compare the results of our algorithm against previous results, which is very interesting. In this sense, case 1 has 5 independent variables and 11 restrictions to meet, case 2 has 5 independent variables

and 11 restrictions, while case 3, the most complete, has 9 independent variables and 13 restrictions to meet at the same time.

### 4.1.1. Case 1: CMOS Differential Amplifier

Figure 5 shows our first case of study, a CMOS differential amplifier, where, $W$ is the width and $L$ is the length of the CMOS transistor. First, $M_1$ must be equally sized than $M_2$; thus, the following equality restrictions must be satisfied:

Secondly, s of the current source, $M_3$ and $M_4$, must be equally sized, too, thus

$$W_3 = W_4 \quad \text{and} \quad L_3 = L_4. \tag{14}$$

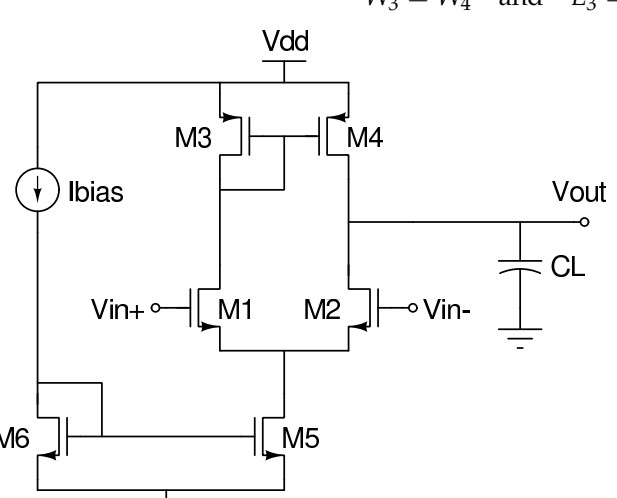

**Figure 5.** CMOS differential amplifier.

$$W_1 = W_2 \quad \text{and} \quad L_1 = L_2. \tag{15}$$

We let both $W_5$ and $W_6$ be independent variables, and our algorithm selects their values while $L_5 = L_6$. That is because the sizes of all s are within a specific range imposed by the technology used for this design:

$$W_{n,min} \leq W_n < W_{n,max}, \quad n = 1, 2, \cdots, 6. \tag{16}$$

In our case, $W_{n,min}$ was fixed to 4 μm for a better comparison with other works, and $W_{n,max}$ was fixed to 120 μm to have a value large enough. For this example, there are 5 independent variables ($W_1$, $W_3$, $W_5$, $W_6$ and $I_{bias}$) and 2 dependent ones ($W_2$ and $W_4$). On the other hand, the design specifications to be met will be treated as constraints. For this case, there are 11 constraints: load capacitance, slew rate, power dissipation, phase margin, cut-off frequency, DC gain, $V_{IC}(min)$, $V_{IC}(max)$, Common Mode Rejection Ratio (CMRR), Positive Power Supply Rejection Ratio (PSRR+) and Negative Power Supply Rejection Ratio (PSRR−).

### 4.1.2. Case 2: CMOS Two-Stage Operational Amplifier

Figure 6 shows our second case of study, a CMOS two-stage operational amplifier consisting of 8 s. The first amplification stage, differential input, has the stipulation that $M_1$ must be equally sized as $M_2$, so that Equations (15) and (14) are still valid, and we add,

$$W_5 = W_8 \quad \text{and} \quad L_5 = L_8. \tag{17}$$

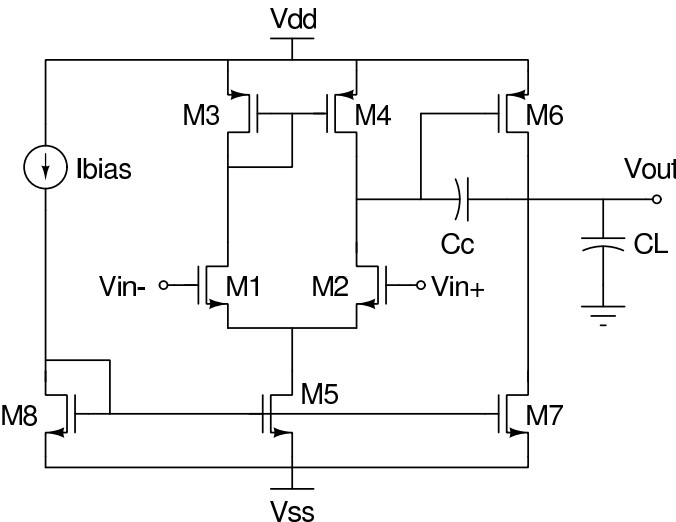

**Figure 6.** CMOS two-stage operational amplifier.

Also, to avoid an output offset at the second amplification stage, the following restriction is imposed:

$$\frac{W_7/L_7}{W_5/L_5} = 2\frac{W_6/L_6}{W_4/L_4}. \tag{18}$$

Similarly, as in (16), sizes of the CMOS two-stage operational amplifier are in a specific range, but now $n = 8$. Also, the compensation capacitance is within a range of values, between $C_{C,min}$ and $C_{C,max}$, which the designer selects:

$$C_{C,min} \leq C_C < C_{C,max}. \tag{19}$$

The $C_{C,min}$ and $C_{C,max}$ values are fed to the Ts-CPD algorithm through a file in our EDA tool. We choose $C_{C,min} = 2$ pF, because lower values than that are challenging to achieve and $C_{C,max} = 14$ pF to avoid using significant areas, but these values are easily changed.

On the other hand, bias current $I_{BIAS}$ also is within a range o values:

$$I_{BIAS,min} \leq I_{BIAS} < I_{BIAS,max}. \tag{20}$$

It is clear from Equations (15), (14) and (17) that, for the purpose of design, $W_2, W_4$ and $W_6$ can be handled as independent variables, while $W_1, W_3$ and $W_5$ as can be handled as dependent ones. $W_7$ is deduced from (18), thus, $W_7$ is also a dependent variable; $I_{BIAS}$ and $C_C$ are considered independent variables whose values are bounded by (19) and (20), respectively. Therefore, this example has 5 independent variables, $W_2, W_4, W_6, I_{BIAS}$ and $C_c$, whose values are selected by our algorithm and 5 dependent variables $W_1, W_3, W_5, W_7$ and $W_8$, whose impact over cost function and restrictions is evaluated by our algorithm to determine new values for independent variables, in an iterative process.

In this paper, the length of s is considered constant. However, when lengths are considered variables, the minimum and maximum values must be established, as for widths in Equation (16). For this case, there are 11 constraints: load capacitance, slew rate, power dissipation, phase margin, unity gain bandwidth, DC gain, $V_{IC}(min)$, $V_{IC}(max)$, CMRR, PSRR+, and PSRR−.

### 4.1.3. Case 3: CMOS Foilded Cascode Operational Transconductance Amplifier

A third case of study is the Folded Cascode Operational Transconductance Amplifier (FCOTA) shown in Figure 7. The transistors $M_1$ and $M_2$ are equally sized; thus, Equation (15) is also valid. We considered the transistor widths $W_3$ and $W_4$ independent variables and $W_5$ and $W_{14}$ dependent ones, as follows:

In addition, $W_6$, $W_8$, $W_{12}$ and $W_{15}$ are considered independent variables while $W_7$, $W_9$, $W_{10}$, $W_{11}$, and $W_{13}$ are considered dependent variables, as follows:

$$W_6 = W_7 = W_{13} \quad \text{and} \quad L_6 = L_7 = L_{13}, \tag{21}$$

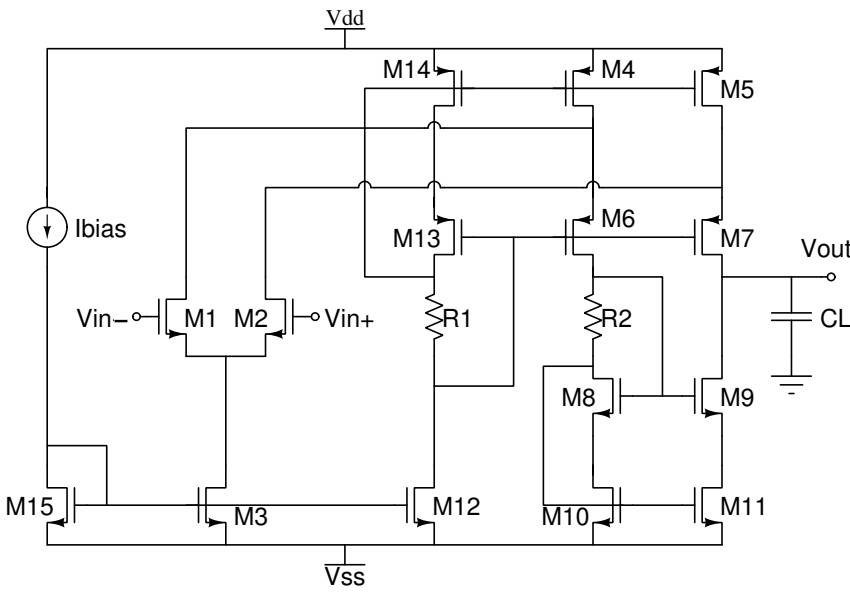

**Figure 7.** CMOS folded cascode operational transconductance amplifier.

$$W_4 = W_5 = W_{14} \quad \text{and} \quad L_4 = L_5 = L_{14}. \tag{22}$$

$$W_8 = W_9 = W_{10} = W_{11} \quad \text{and} \quad L_8 = L_9 = L_{10} = L_{11}. \tag{23}$$

**Table 5.** Design criteria for CMOS differential amplifier (Case 1) and results obtained with several evolutionary algorithms. The best values are in bold.

| Design Criteria | Specs. | Ts-CPD | MOL [62] | SOA [63] | PSO [22] | HS [45] | DE [45] | ABC [45] | GA [44] |
|---|---|---|---|---|---|---|---|---|---|
| Load capacitance (pF) | $\geq 2$ | 2.1 | 5 | 3.5 | 5 | 5 | 5 | 5 | 2 |
| Slew rate (V/µs) | $\geq 10$ | 24.3 | 10 | 12.28 | 22.4 | 14.916 | 18.451 | 15.67 | 3.2 |
| Power dissipation (µW) | $\leq 2000$ | 1075 | 863 | 117 | 1260 | 886 | 990 | 830 | 31 |
| Phase margin (°) | $>45$ | 86.1 | 89 | 83.73 | 83.8 | 89.1 | 88.81 | 91.248 | 72 |
| Cut-off frequency (KHz) | $\geq 100$ | 100.5 | - | 104.8 | 100 | 114 | 129.7 | 112.367 | - |
| Unity gain bandwidth (MHz) | $\geq 1$ | 10 | 17.87 | 12.5 | 12.3 | - | - | - | 3.8 |
| DC gain (dB) | $\geq 40$ | 40.3 | 30 | 44.02 | 42 | 40.98 | 41.23 | 42.045 | 60 |
| $V_{IC}(\text{min})$ (V) | $\geq -1.5$ | −0.8 | −0.5 | −0.37 | −0.8 | −0.7 | −0.92 | −0.97 | −1.3 |
| $V_{IC}(\text{max})$ (V) | $\leq 2$ | 1.1 | 0.7 | 1.57 | 1.4 | 1.2 | 1.15 | 1.2 | 1.9 |
| CMRR (dB) | $>40$ | 81.0 | 59 | 83.17 | 84.2 | 78.5 | 78.39 | 79.67 | - |
| PSRR+ (dB) | $>40$ | 41.2 | 41 | 60.59 | 40.1 | 42.93 | 43.14 | 43.857 | - |
| PSRR− (dB) | $>40$ | 78.1 | 68 | 108.6 | 68 | 67.64 | 68.175 | 68.423 | - |
| Total component area (µm²) | $<300$ | **109** | 235 | 236 | 296 | - | - | - | 6500 |
| AFOM$_{SS}$ (MHz·pF)/(µW·mm²) | | 179 | **457** | 318 | 165 | - | - | - | 40 |

The values of the bias current $I_{BIAS}$ are bounded by (20) and properly selected by our algorithm. For design, we considered $R_1$ as an independent variable. Thus, our algorithm also selects its value within $1000 < R_1 < 6000$, while $R_2$ is considered a dependent variable, with $R_1 = R_2$. This way, there are 9 independent variables ($W_1$, $W_3$, $W_4$, $W_6$, $W_8$, $W_{12}$, $W_{15}$, $I_{BIAS}$ and $R_1$) and 9 dependent variables ($W_2$, $W_5$, $W_7$, $W_9$, $W_{10}$, $W_{11}$, $W_{13}$, $W_{14}$ and $R_2$). The constraints for this case are 13: load capacitance, slew rate, power dissipation, phase margin, unity gain bandwidth, DC gain, $V_{IC}(\text{min})$, $V_{IC}(\text{max})$, $V_{out}(\text{min})$, $V_{out}(\text{max})$, CMRR, PSRR+ and PSRR−.

## 5. Numerical Results and Discussion

In order to test our proposed tool, three examples of design are shown in this section. First, the optimization is implemented in MATLAB R2014b, while the simulation of circuits is implemented in the NGSPICE r26 simulator; both are linked, so the design process is completely automated. On the other hand, the model of NMOS and PMOS transistors for 0.35 μm technology was downloaded from the MOL database. Finally, the transistor lengths were set to fixed values close to those in the literature for comparison purposes.

Our design objective is to minimize the area of analog circuits. However, designing an amplifier is always a trade-off, so we introduce the Area Figure of Merit for small-signal ($AFOM_{ss}$) that considers silicon area to assess the designed circuits' overall performance [64]:

$$AFOM_{SS} = (f_u \cdot C_L / P^Q \cdot Area) \tag{24}$$

where $f_u$ is the unity gain frequency, $C_L$ is the load capacitance, $P^Q$ is the power consumption at quiescent, and Area is the component (transistors) area.

### 5.1. Numerical Results for CMOS Differential Amplifier (Case 1)

As a first example, the differential amplifier of Figure 5 is designed. We aim to minimize the total component area, which is our cost function, below 300 μm$^2$ while restrictions are still met. As shown in Table 5, the power dissipation is specified to be <2200 μW, DC gain ≥40 dB, slew rate ≥ 10 V/μs and the cut-off frequency ≥ 100 KHz. Other specifications are CMRR, PSRR+, PSRR−, and the Input Common-Mode Range (ICMR), all to be >40 dB, and finally $V_{IC(min)} \geq -1.5$ V and $V_{IC(max)} \leq 2$ V. The circuit's load determines load capacitance, but the specification to be satisfied is ≥2 pF; we choose 2.1 pF. The AFOM$_{SS}$ is also shown.

For the optimization purpose, some variables are set to a fixed value, and the microchannel lengths were set to $L_1 = L_2 = L_3 = L_4 = 3.5$ μm, $L_5 = L_6 = 1.4$ μm, and voltage sources were set to $V_{dd} = -V_{ss} = 2.5$V. On the other hand, $C_c$ and $I_{bias}$ are treated as independent variables with restrictions, i.e., they can run within a specific range of values in our algorithm.

The numerical results for the differential amplifier of Figure 5 are shown in Table 5; it presents a comparison of Ts-CPD with several methods: Many Optimizing Liaisons (MOLs) [62], Seeker Optimization Algorithm (SOA) [63], PSO [22], Harmony Search (HS) [45], DE [45], Artificial Bee Colony (ABC) [45], and GA [44]. The Ts-CPD obtains the lower total component area for methods that report this design objective and obtains the higher slew rate and PSRR−; other specifications are also accomplished. Here, the MOLs algorithm has the higher AFOM$_{SS}$ value. Table 6 shows the result of the designed differential amplifier for three evolutionary algorithms.

**Table 6.** Design parameters for three algorithms (Case 1).

| Design Parameters | Ts-CPD | PSO [22] | GA [44] |
|---|---|---|---|
| $W_1/L_1$ (μm/μm) | 7.6/3.5 | 29.4/3.5 | 240/13.2 |
| $W_2/L_2$ (μm/μm) | 7.6/3.5 | 29.4/3.5 | 240/13.2 |
| $W_3/L_3$ (μm/μm) | 4.6/3.5 | 11.3/3.5 | 7.3/7.7 |
| $W_4/L_4$ (μm/μm) | 4.6/3.5 | 11.3/3.5 | 7.3/7.7 |
| $W_5/L_5$ (μm/μm) | 5.9/1.4 | 4.2/1.4 | 4.6/2.4 |
| $W_6/L_6$ (μm/μm) | 11.2/1.4 | 4.2/1.4 | 2.4/2.4 |
| $I_{bias}$ (μA) | 141 | 125 | 2 |

In order to explore the performance of the differential amplifier designed, we show the DC gain and phase margin in Figure 8a; The CMRR, PSRR+, and PSRR− in Figure 8b; Slew rate in Figure 8c; and the ICMR in Figure 8d, which is used for the graphical determination of $V_{IC}$(min) and $V_{IC}$(max). These graphics demonstrate that the designed circuit behaves well and is accomplished with all the constraints (Specifications).

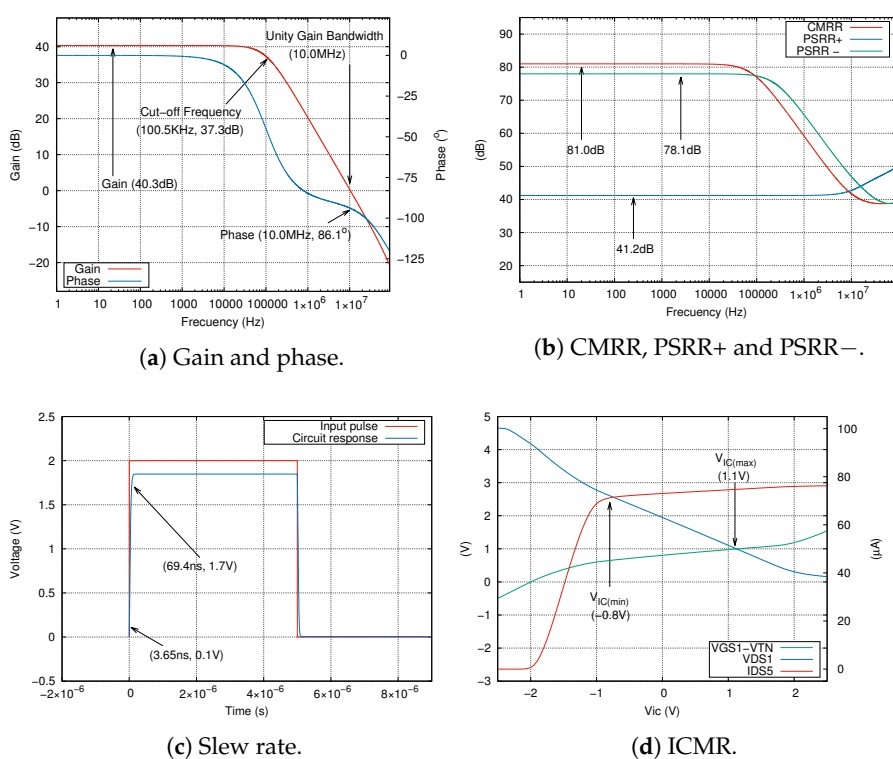

(**a**) Gain and phase.

(**b**) CMRR, PSRR+ and PSRR−.

(**c**) Slew rate.

(**d**) ICMR.

**Figure 8.** Performance of CMOS differential amplifier.

Figure 9a shows the convergence of our algorithm for this circuit design, which has an excellent profile. Our algorithm's behavior was also tested with 50 runs; the corresponding Box and Whisker plot is shown in Figure 9b. The median is $1.4168 \times 10^{-10}$ m$^2$, which is still below the results reported for other algorithms; see Table 5.

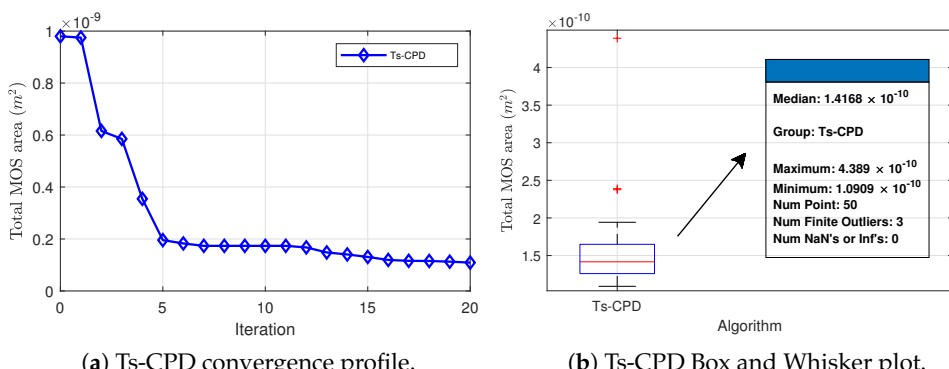

(**a**) Ts-CPD convergence profile.

(**b**) Ts-CPD Box and Whisker plot.

**Figure 9.** Ts-CPDtest for CMOS differential amplifier.

### 5.2. Numerical Results for CMOS Two-Stage Operational Amplifier (Case 2)

As a second example, we designed the two-stage operational amplifier in Figure 6. Again, the aim is to minimize the total component area as much as possible while constraints are still met. The total component area is specified to be $<300$ μm$^2$, and in this case, the DC gain $> 60$ dB, unity gain bandwidth $\geq 3$ MHz, phase margin $\geq 45°$, slew rate $\geq 10$ V/μs and load capacitance $\geq 7$ pF. In other set of specifications, CMRR$> 60$ dB, PSRR$^+ > 70$ dB, PSRR$^+ > 70$ dB, $V_{IC(min)} > -1.5$ V and $V_{IC(max)} \leq 2$ V. At the end, the AFOM$_{SS}$ is shown.

The microchannel lengths of all MOS transistors have been set to a fixed value, $L_1 = L_2 \cdots L_8 = 0.8$ μm, while voltage sources are set to $V_{dd} = -V_{ss} = 2.5$ V. Here, $C_c$ and $I_{bias}$ are independent variables. Thus, our algorithm determines its values in concordance with (19) and (20), respectively.

Table 7 shows the complete set of restrictions and design objective for the CMOS operational amplifier of Figure 6, as well as the comparison of methods Ts-CPD, GSA-PSO [23], PSO, and Geometric Programming (GP) [46]. As expected, the Ts-CPD has the lower component area and the highest slew rate and PSRR−. The AFOM$_{SS}$, on the other hand, is higher for our algorithm. The design parameters of the optimized circuit are shown in Table 8.

**Table 7.** Design criteria for CMOS two-stage operational amplifier and results obtained with several algorithms. The best values are in bold.

| Design Criteria | Specs. | Ts-CPD | GSA-PSO [23] | PSO [22] | GP [46] |
|---|---|---|---|---|---|
| Load capacitance (pF) | ≥7 | 7.1 | 7.2 | 10 | 3 |
| Slew rate (V/µs) | ≥10 | 11.9 | 10.88 | 11.13 | 88 |
| Power dissipation (µW) | ≤2500 | 1084 | 712.8 | 2370 | 5000 |
| Phase margin (°) | >45 | 46 | 66.2 | 66.55 | 60 |
| Unity gain bandwidth (MHz) | ≥3 | 6.2 | 5.776 | 5.32 | 86 |
| DC gain (dB) | >60 | 64.7 | 75.43 | 63.8 | 89.2 |
| $V_{IC}$(min) (V) | ≥−1.5 | −1.15 | −0.886 | −0.8 | - |
| $V_{IC}$(max) (V) | ≤2 | 1.6 | 1.9 | 1.75 | - |
| CMRR (dB) | >60 | 74.0 | 75.43 | 63.8 | 89.2 |
| PSRR+ (dB) | >70 | 72.5 | 83.2 | 78.27 | 116 |
| PSRR− (dB) | >70 | 92.9 | 110.4 | 93.56 | 98.4 |
| Total component area (µm²) | <300 | **45.6** | 109.6 | 265 | 8200 |
| AFOM$_{SS}$ (MHz·pF)/(µW·mm²) | | **902** | 532 | 85 | 6 |

**Table 8.** Design parameters for the four algorithms (Case 2).

| Design Variables | Ts-CPD | GSA-PSO [23] | PSO [22] | GP [46] |
|---|---|---|---|---|
| $W_1/L_1$ (µm/µm) | 4.1/0.8 | 4/2 | 4.9/2 | 232.8/0.8 |
| $W_2/L_2$ (µm/µm) | 4.1/0.8 | 4/2 | 4.9/2 | 232.8/0.8 |
| $W_3/L_3$ (µm/µm) | 4.0/0.8 | 4/2 | 5.9/2 | 143.6/0.8 |
| $W_4/L_4$ (µm/µm) | 4.0/0.8 | 4/2 | 5.9/2 | 143.6/0.8 |
| $W_5/L_5$ (µm/µm) | 4.7/0.8 | 2.8/2 | 2.1/2 | 64.6/0.8 |
| $W_6/L_6$ (µm/µm) | 19.8/0.8 | 24/2 | 90.9/2 | 588.8/0.8 |
| $W_7/L_7$ (µm/µm) | 11.5/0.8 | 9.2/2 | 16.3/2 | 132.6/0.8 |
| $W_8/L_8$ (µm/µm) | 4.7/0.8 | 2.8/2 | 2.1/2 | 2/0.8 |
| $C_C$ (pF) | 3.8 | 2.8 | 3 | 3.5 |
| $I_{bias}$ (µA) | 42.7 | 28 | 40.39 | 10 |

The performance of the CMOS two-stage operational amplifier can be evaluated through the gain and phase plot in Figure 10a; the CMRR, PSRR+, and PSRR− plots in Figure 10b; the ICMR in Figure 10c; and the slew rate in Figure 10d. These plots also demonstrate the excellent performance of the designed circuit.

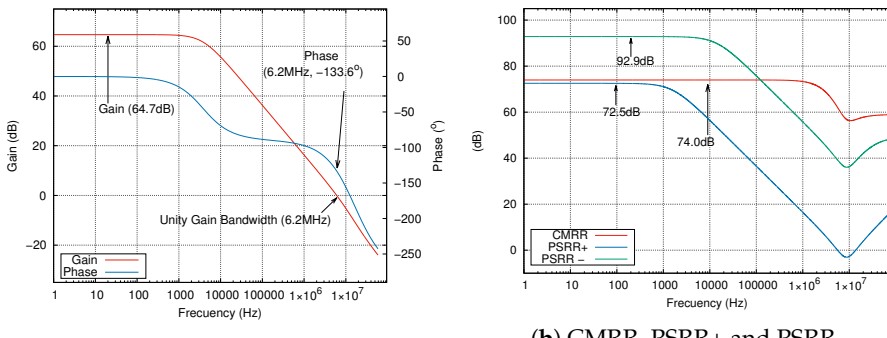

**(a)** Gain and phase.

**(b)** CMRR, PSRR+ and PSRR−.

**Figure 10.** *Cont.*

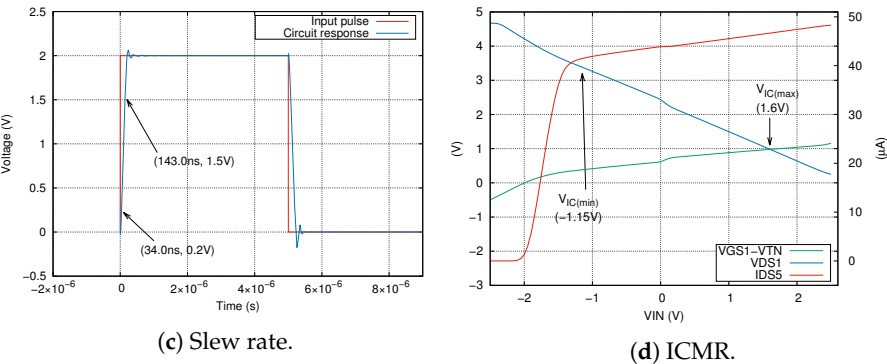

(**c**) Slew rate.

(**d**) ICMR.

**Figure 10.** Performance of CMOS two-stage operational amplifier.

On the other hand, we evaluated the performance of our algorithm with the convergence profile in Figure 11a, and the Box and Whisker plot of Figure 11b. After 16 iterations, the Ts-CPD reached convergence; see Figure 11a. We executed 50 trial runs for the circuit design; Figure 11b shows the corresponding Box and Whisker plot for the total MOS area of transistors. The best value is $4.557 \times 10^{-11}$ m$^2$, but the median ($6.1738 \times 10^{-11}$ m$^2$) is also lower than others reported for this circuit, as can be seen in Table 7.

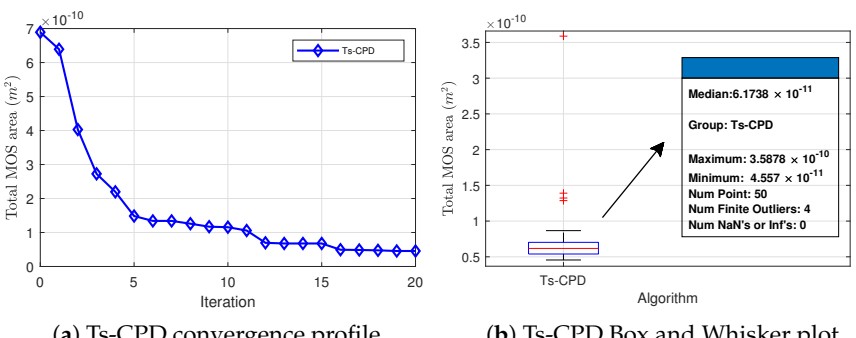

(**a**) Ts-CPD convergence profile.

(**b**) Ts-CPD Box and Whisker plot.

**Figure 11.** Ts-CPD test for CMOS two-stage operational amplifier.

*5.3. Numerical Results for CMOS Folded Cascode Operational Transconductance Amplifier (Case 3)*

Our third example is the folded cascode operational amplifier shown in Figure 7. The total component area specified is $< 1315.9$ μm$^2$ (our design objective). At the same time, specified constraints are gain $> 74$ dB, unity bandwidth $\geq 10$ MHz, phase margin $> 60°$, slew rate $\geq 10$ V/μs and load capacitance $\geq 10$ pF (we chose exactly 10.0 pF). More constraints are CMRR, PSRR+, PSRR− all three $\geq 55$ dB, $V_{IC(min)} \geq -1.5$, $V_{IC(max)} \leq 2.5$, and finally $V_{out(min)} \geq -2$ and $V_{out(max)} \leq 2$. And at the end, the AFOM$_{SS}$ is shown.

For all MOS transistors, the lengths have been set to a fixed value, $L_1 = L_2 \cdots$ $L_{15} = 1.5$ μm, and the voltage sources are set to $V_{dd} = -V_{ss} = 2.5$ V. Besides the transistor widths ($W_i$), $I_{bias}$, $R_1$ and $R_2$ are also variables.

Table 9 shows the numerical results for the FCOTA of Figure 7 and a comparison of methods Ts-CPD and PSO with Aging Leader and Challengers (ALC-PSO) [1]. Our proposal, Ts-CPD, has the lower total component area (our design objective) and the highest Unity gain bandwidth, phase margin, CMRR, and PSRR−, while other constraints are also met. Additionally, the AFOM$_{SS}$ is greater for our algorithm. The parameters of the optimized circuit for the two proposals are shown in Table 10.

**Table 9.** Designcriteria for CMOS folded cascode operational transconductance amplifier. The best values are in bold.

| Design Criteria | Specs. | Ts-CPD | ALC-PSO [1] |
|---|---|---|---|
| Load capacitance (pF) | $\geq$10 | 10.0 | 10.028 |
| Slew rate (V/$\mu$s) | $\geq$10 | 13.8 | 19.37 |
| Power dissipation (m$W$) | $\leq$5 | 3.3 | 2.504 |
| Phase margin ($^\circ$) | >60 | 83.9 | 63.1 |
| Unity gain bandwidth (MHz) | $\geq$10 | 17.8 | 11.11 |
| DC gain (dB) | >74 | 74.1 | 76.97 |
| $V_{IC}$(min) (V) | $\geq$$-$1.5 | $-$0.69 | $-$1.466 |
| $V_{IC}$(max) (V) | $\leq$2.5 | 2.41 | 2.486 |
| $V_{out(min)}$ (V) | $\geq$$-$2 | $-$2.0 | $-$1.936 |
| $V_{out(max)}$ (V) | $\leq$2 | 1.99 | 1.996 |
| CMRR (dB) | >55 | 111.8 | 87.58 |
| PSRR+ (dB) | >55 | 82.9 | 84.21 |
| PSRR$-$ (dB) | >55 | 74.6 | 61.47 |
| Total component area ($\mu$m$^2$) | <1315.9 | **600.9** | 835.2625 |
| AFOM$_{SS}$ (MHz$\cdot$pF)/($\mu$W$\cdot$mm$^2$) | | **89,764** | 53,269 |

**Table 10.** Design parameters for Case 3.

| Design Variables | Ts-CPD | ALC-PSO [1] |
|---|---|---|
| $W_1/L_1$ ($\mu$m/$\mu$m) | 48.43/1.25 | 60.46/1.25 |
| $W_2/L_2$ ($\mu$m/$\mu$m) | 48.43/1.25 | 60.46/1.25 |
| $W_3/L_3$ ($\mu$m/$\mu$m) | 78.66/1.25 | 35.8/1.25 |
| $W_4/L_4$ ($\mu$m/$\mu$m) | 13.40/1.25 | 40.1/1.25 |
| $W_5/L_5$ ($\mu$m/$\mu$m) | 13.40/1.25 | 40.1/1.25 |
| $W_6/L_6$ ($\mu$m/$\mu$m) | 24.26/1.25 | 45.94/1.25 |
| $W_7/L_7$ ($\mu$m/$\mu$m) | 24.26/1.25 | 45.1/1.25 |
| $W_8/L_8$ ($\mu$m/$\mu$m) | 25.35/1.25 | 59.63/1.25 |
| $W_9/L_9$ ($\mu$m/$\mu$m) | 25.35/1.25 | 59.63/1.25 |
| $W_{10}/L_{10}$ ($\mu$m/$\mu$m) | 25.35/1.25 | 59.63/1.25 |
| $W_{11}/L_{11}$ ($\mu$m/$\mu$m) | 25.35/1.25 | 59.63/1.25 |
| $W_{12}/L_{12}$ ($\mu$m/$\mu$m) | 55.60/1.25 | 14.85/1.25 |
| $W_{13}/L_{13}$ ($\mu$m/$\mu$m) | 24.26/1.25 | 45.94/1.25 |
| $W_{14}/L_{14}$ ($\mu$m/$\mu$m) | 13.34/1.25 | 40.1/1.25 |
| $W_{15}/L_{15}$ ($\mu$m/$\mu$m) | 35.23/1.25 | - |
| $I_{bias}$ ($\mu$A) | 119.3 | - |
| $R_1$ (k$\Omega$) | 4.83 | 1.89 |
| $R_2$ (k$\Omega$) | 4.83 | 1.89 |

The excellent performance of the CMOS folded cascode operational transconductance amplifier is demonstrated through the plots of gain and phase in Figure 12a; CMRR, PSRR+, and PSRR$-$ in Figure 12b; the slew rate in Figure 12c; and the ICMR, Figure 12d.

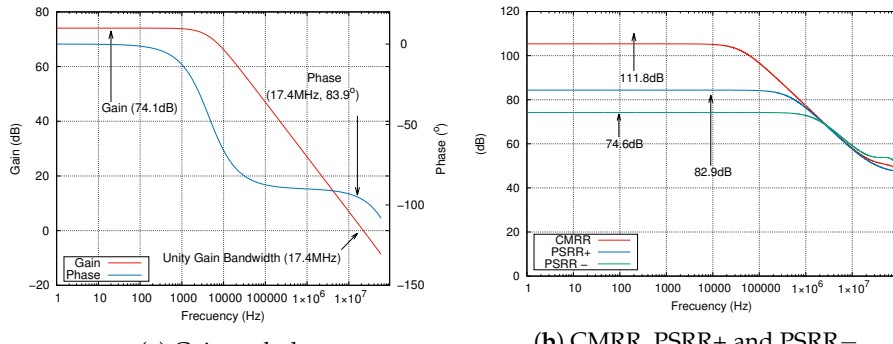

(**a**) Gain and phase.

(**b**) CMRR, PSRR+ and PSRR$-$.

**Figure 12.** *Cont.*

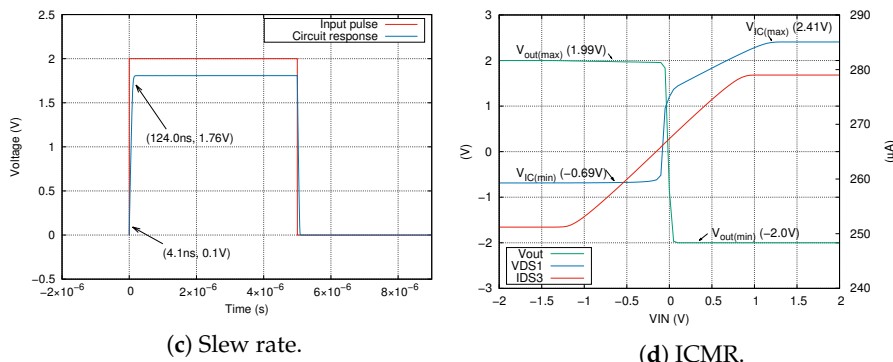

(**c**) Slew rate.                                                    (**d**) ICMR.

**Figure 12.** Performance of CMOS folded cascode operational transconductance amplifier.

The Ts-CPD performance is evaluated with the convergence profile shown in Figure 13a and the Box and Whisker plot of Figure 13b. As can be seen in Figure 13a, the Ts-CPD converges very quickly for this circuit design in just 5 iterations. Figure 13b shows the Box and Whisker plot for 50 trial runs for the total MOS area of transistors. The median is $5.9674 \times 10^{-11}$ $m^2$, and the solutions are very clustered towards this value.

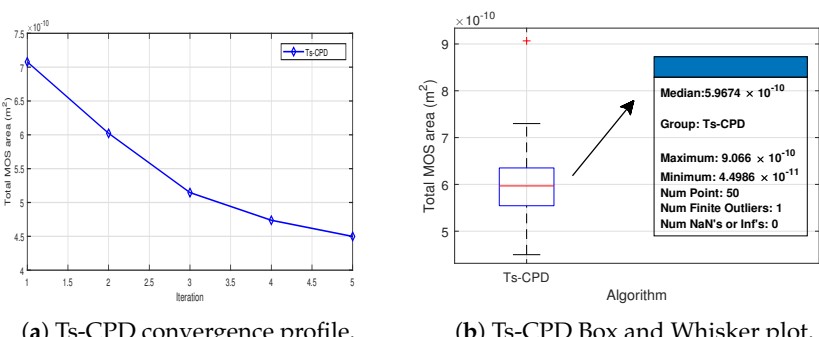

(**a**) Ts-CPD convergence profile.                        (**b**) Ts-CPD Box and Whisker plot.

**Figure 13.** Ts-CPD test for CMOS folded cascode operational transconductance amplifier.

## 6. Conclusions

The Ts-CPSO algorithm that was proposed and implemented improves the CPSO by incorporating a way of evaluating the performance of constraints, through the optimization-with-constraints method, with a new rule we proposed. This algorithm has the advantage of not only minimizing the objective function but also ensuring that the constraints are met and then generating the new parameter values. Then the Ts-CPSO algorithm is incorporated into our EDA tool for the optimal sizing of analog circuits, which does not require mathematical equations since the optimization is linked to a simulator that provides the circuit's behavior.

The Ts-CPD algorithm, as part of our EDA tool, was tested with three cases of study in a 0.35 μm CMOS technology, a differential amplifier, a two-stage operational amplifier, and a folded cascode operational transconductance amplifier. It was proposed as a design objective to reduce the total area occupied by the transistors while complying with some established constraints. In all cases, our tool found a better solution, for the objective, than previously reported tools, while the constraints were kept within the desired limits.

In future work, we are going to implement a multi-objective algorithm, which we will add as the kernel of our EDA tool. We will also do design tests with analog circuits with more transistors and large-scale analog circuits, such as the Analog-to-Digital Converter (ADC), considering the Layout design. As another potential future project, a framework incorporating multiple algorithms for optimizing various analog circuits can be developed. This framework would allow users to customize each algorithm's parameters to enhance its performance, compare the different methods with convergence plots and identify the optimal design. It would be interesting to conduct a future study comparing Ts-CPD with

other algorithms that are known for their success in solving CEC test problems and real-world applications. Some of these algorithms include Adaptive Differential Evolution with Optional External Archive (JADE), Success-History Based Adaptive Differential Evolution (SHADE), Self-adaptive Differential Evolution with Lévy-flight (LSHADE) and Improving Multi-objective Differential Evolutionary (IMODE).

**Author Contributions:** Conceptualization, P.L.-E. and P.M.-R.; methodology, P.L.-E. and P.M.-R.; validation, J.C.S.-T.-M. and N.H.-R.; formal analysis, P.L.-E., P.M.-R. and J.C.S.-T.-M.; investigation, P.L.-E., P.M.-R., J.C.S.-T.-M. and N.H.-R.; resources, P.M.-R. and J.C.S.-T.-M.; writing—original draft preparation, P.L.-E., P.M.-R., J.C.S.-T.-M. and N.H.-R.; writing—review and editing, P.M.-R. and J.C.S.-T.-M. visualization, P.L.-E., P.M.-R. and J.C.S.-T.-M.; supervision, J.C.S.-T.-M. and N.H.-R.; funding acquisition, J.C.S.-T.-M. All authors have read and agreed to the published version of the manuscript.

**Funding:** This study was supported by the Autonomous University of Hidalgo (UAEH) and the National Council for Humanities, Science and Technology (CONAHCYT) with project number F003-320109.

**Institutional Review Board Statement:** Not applicable.

**Informed Consent Statement:** Not applicable.

**Data Availability Statement:** The Ts-CPD source code is available on Github https://github.com/pmirandar/Ts-CPD-EDA-Tool (accessed on 20 October 2023).

**Conflicts of Interest:** The authors declare that they have no known competing financial interest or personal relationships that could have appeared to influence the work reported in this paper.

## Abbreviations

The following abbreviations were used in this research:

| | |
|---|---|
| ABC | Artificial Bee Colony |
| ACO | Ant Colony Optimization |
| ADC | Analog-to-Digital Converter |
| $AFOM_{SS}$ | Area Figure of Merit (for small signal) |
| ALC-PSO | PSO with Aging Leader and Challengers |
| AOA | Archimedes Optimization Algorithm |
| CA | Cellular Automata |
| CAD | Computer-Aided Design |
| CCAA | Continuous-state Cellular Automata Algorithm |
| CGA | Customized Genetic Algorithm |
| CMOS | Complementary Metal-Oxide-Semiconductor |
| CMRR | Common Mode Rejection Ratio |
| CPSO-DE | Cellular Particle Swarm Optimization with Differential Evolution |
| CRPSO | Crazy PSO |
| DE | Differential Evolution |
| ECA | Elementary Cellular Automaton |
| EDA | Electronic Design Automation |
| FCOTA | Folded Cascode Operational Transconductance Amplifier |
| GA | Genetic Algorithm |
| GP | Geometric Programming |
| GSA-PSO | Gravitational Search Algorithm with PSO |
| HHO | Harris Hawks Optimization |
| HS | Harmony Search |
| ICMR | Input Common-Mode Range |
| LCPSO | Leader and Challenger PSO |
| MmCCAA | Majority-minority Cellular Automata Algorithm |
| MOLs | Many Optimizing Liaisons |
| MOPSO | Multi-Objective Particle Swarm Optimization |
| MOS | Metal-Oxide-Semiconductor |
| MOSA | Multi-Objective Simulated Annealing |
| NLP | Nonlinear Programming |
| NMOS | N-Channel MOS |
| NSGA | Non-dominated Sorting Genetic Algorithm |
| op-amp | Operational Amplifier |
| PMOS | P-Channel MOS |

| PO | Political Optimizer |
|------|------|
| PSO | Particle Swarm Optimization |
| PSRR | Power Supply Rejection Ratio |
| PSRR+ | Positive Power Supply Rejection Ratio |
| PSRR− | Negative Power Supply Rejection Ratio |
| RECAA | Reversible Elementary Cellular Automata |
| SA | Simulated Annealing |
| SOA | Seeker Optimization Algorithm |
| Ts-CDP | Tournament-selection CPSO-DE |
| VLSI | Very Large Scale of Integration |
| WSA | Weighted Superposition Attraction |

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
