# Peer review of "Improvement in Sizing Constrained Analog IC via Ts-CPD Algorithm"

_computation, doi:10.3390/computation11110230_

Round 1
Reviewer 1 Report
Comments and Suggestions for Authors
I recommend accepting this paper after minor revision.
1) The Introduction can be further streamlined by removing some descriptions that are not related to this article.
2) The original data, code, and experimental results of the paper should be made public to all readers, such as Github.
Author Response
Kindly review the attached file.
Thank you.

Reviewer 2 Report
Comments and Suggestions for Authors
This paper proposes an algorithm to improve the cellular particle swarm optimization process through the optimization-with-constraints method. This approach is tested with different cases of study to demonstrate its performance. The quality of the paper can be further improved with more theoretical analysis of the approach.
Author Response

(The authors gave the same response as above.)

Reviewer 3 Report
Comments and Suggestions for Authors
The paper introduces a variation of the cellular particle swarm optimization algorithm with differential evolution hybridization, implemented as a kernel in an electronic design automation (EDA) tool for sizing analog circuit components. The paper and proposed approach are interesting. The paper can be accepted after addressing the following comments.
Major comments:
1- The authors should check the manuscript thoroughly to provide and correctly use the acronyms in the right place. For instance, GP is first used on page 2 but introduced on page 3. CMOS is used on page 2 or CCAA on page 3 but has yet to be introduced. There are many other instances. The paper should be readable by non-expert readers as well.
2- Although many references are cited in the manuscript, there is a notable lack of references from the same journal in which the manuscript has been submitted. I suggest the authors cite and discuss related references from the Computation journal in the Introduction section.
Minor comments:
1- Abstract: “to include constrained optimization in it, named” → “to include constrained optimization, named”.
Author Response

(The authors gave the same response as above.)

Reviewer 4 Report
Comments and Suggestions for Authors
1. First and foremost, I must emphasize that I do not agree with the author's assertion about the PSO based on cellular automata. I believe the author merely generated a few new entities around the original one for local search, without employing any mechanisms related to the reproduction or apoptosis of cellular automata. This approach to writing hints at sensationalism and exaggeration. The author must provide a valid explanation to convince me; otherwise, they should revise the descriptions in the paper.
2. Not only did the author use local search, but they also employed differential operations on top of the PSO operations. This will significantly increase the algorithm's complexity. Therefore, the author needs to provide a complexity analysis to prove that the performance improvement of the algorithm outweighs the downside of requiring substantial computational resources.
3. The algorithms the author compared with are evidently classical ones. However, a series of newer algorithms like JADE, SHADE, LSHADE, and IMODE clearly offer enhanced performance. If the author does not make these comparisons, it will significantly undermine the credibility of this paper. However, given that the complexity of the new model proposed by the author is unknown, if modifications cannot be completed this time, please compare with these algorithms in subsequent research.
4. The author needs to improve in terms of paper writing. For instance, the "h(G)" mentioned in line 180 was not explained in advance.
5. In summary, I believe the author has expanded the application scope of metaheuristic algorithms, and their approach to studying classic algorithms like PSO and DE is much superior to the so-called "zoo algorithms". Thus, it is commendable. However, regarding the issue of introducing new concepts like cellular automata without proper justification, the author needs to provide a reasonable explanation.
Author Response

(The authors gave the same response as above.)

Round 2
Reviewer 3 Report
Comments and Suggestions for Authors
The authors have addressed my comments well.
Author Response
Please, kindly review the attached file.

Reviewer 4 Report
Comments and Suggestions for Authors
Regarding the section on cellular automata, I still find it somewhat forced. Therefore, I hope the author will focus more on the technology itself in future research, rather than applying famous concepts.
Author Response

(The authors gave the same response as above.)
